

# Use of an ultra-long-range terrestrial laser scanner to monitor the mass balance of very small glaciers in the Swiss Alps

Mauro Fischer[1], Matthias Huss[1,2], Mario Kummert[1], and Martin Hoelzle[1]

[1]Department of Geosciences, University of Fribourg, 1700 Fribourg, Switzerland
[2]Laboratory of Hydraulics, Hydrology and Glaciology (VAW), ETH Zurich, 8093 Zurich, Switzerland

*Correspondence to:* Mauro Fischer (mauro.fischer@unifr.ch)

**Abstract.** Due to the relative lack of empirical field data, the response of very small glaciers ($<0.5\,\text{km}^2$) to current atmospheric warming is not fully understood yet. Investigating their mass balance is a prerequisite to fill this knowledge gap. Application of the direct glaciological method is one option. Since most recently, terrestrial laser scanning (TLS) techniques operating in the near infrared range are successfully applied for the creation of repeated high-resolution digital elevation models and

consecutive derivation of annual geodetic mass balances of very small glaciers. This method is promising, as laborious and potentially dangerous field measurements as well as the inter- and extrapolation of point measurements can be circumvented. However, it still owes to be validated. Here, we present TLS-derived annual surface elevation and geodetic mass changes for five very small glaciers in Switzerland (Glacier de Prapio, Glacier du Sex Rouge, St. Annafirn, Schwarzbachfirn, and Pizolgletscher) and two consecutive years (2013/14–2014/15). The scans were acquired with an ultra-long-range *Riegl* VZ-

6000 especially designed for surveying snow- and ice-covered terrain. Zonally variable conversion factors for firn and bare ice surfaces were applied to convert geodetic volume to mass changes. We compare the geodetic results to direct glaciological mass balance measurements coinciding with the TLS surveys and carefully assess the uncertainties and errors included in both methods. Average glacier-wide mass balances were negative in both years, showing remarkably stronger mass losses in 2014/15 (–1.65 m w.e.) compared to 2013/14 (–0.59 m w.e.). Geodetic mass balances were slightly less negative but in close

agreement with the direct glaciological ones ($R^2 = 0.91$). Due to the very dense in-situ measurements, the uncertainties in the direct glaciological mass balances were small compared to the majority of measured glaciers worldwide ($\pm 0.09\,\text{m w.e. yr}^{-1}$ on average), and similar to uncertainties in the TLS-derived geodetic mass balances ($\pm 0.13\,\text{m w.e. yr}^{-1}$).

## 1 Introduction

Around 80% of the *number* of glaciers in the European Alps (Fischer et al., 2014; Gardent et al., 2014; Fischer et al., 2015a;

Smiraglia et al., 2015), and in mid- to low-latitude mountain ranges in general (Pfeffer et al., 2014), are smaller than $0.5\,\text{km}^2$ and hence belong to the size class of very small glaciers (Huss, 2010). Despite their predominance in absolute number, very small glaciers have always received little attention in glaciological research, and empirical knowledge is mainly based on





studies focusing on the Mediterranean Mountains (e.g., Grunewald and Scheithauer, 2010). However, due to their vast number and short response time, very small glaciers are even relevant at larger scales, as they impact on the hydrology of certain catchments (Jost et al., 2012) and, at least over the next one or two decades, even notably contribute to global sea-level rise (Huss and Hock, 2015). Even if an increasing interest in very small glaciers of the European Alps could be observed since

most recent years (e.g., Hagg et al., 2008; Huss, 2010; Carturan et al., 2013; Gilbert et al., 2012; Bosson et al., 2014), field measurements are still sparse, especially for the central Alps, meaning that there is considerable uncertainty in the response of very small glaciers to atmospheric warming.

Measuring glacier mass balance is important to understand the glacier-climate interaction as it directly reflects the climatic forcing on the glacier (e.g., Vincent et al., 2004). In contrast to annual field measurements on individual glaciers using the

direct glaciological method (Østrem and Brugman, 1991; Cogley et al., 2011), mass balance can also be reconstructed from the comparison of two different digital elevation models (DEMs) of the glacier surface topography, the so-called geodetic method (e.g., Rignot et al., 2003; Zemp et al., 2013). Until a few years ago, the accuracy of such DEMs, for instance derived from photogrammetry, mostly limited the time resolution of reliable geodetic mass balance measurements to a multi-annual to decadal scale (Cox and March, 2004; Thibert et al., 2008). Today, Light Detection And Ranging (LiDAR) techniques from

aircraft, so-called airborne laser scanning (ALS), allows derivation of dense point clouds and hence creation of high-resolution DEMs over snow and ice and the computation of glacier surface elevation, volume and geodetic mass changes on an annual or semi-annual basis (Arnold et al., 2006; Joerg et al., 2012; Colucci et al., 2014; Helfricht et al., 2014; Piermattei et al., 2015a).

Even though the initial costs of the scanner and software license are high, terrestrial laser scanning (TLS) techniques are easier and more cost-efficiently applied to individual sites and on the annual to seasonal timescale compared to ALS techniques

(Heritage and Large, 2009). As often nearly the entire surface of very small glaciers is visible from one single location (e.g., from a frontal moraine, an accessible mountain crest or summit, or from the opposite valley side), TLS is a highly promising technique to generate repeated high-resolution DEMs, as well as to derive annual geodetic mass balances of very small glaciers. Moreover, it may have the potential to circumvent laborious and time-consuming in-situ measurements, and to avoid the spatial inter- and extrapolation of point measurements over the entire glacier surface, which is known as an important source of

uncertainty in direct glaciological mass balances (Zemp et al., 2013).

Since 2000, TLS has evolved into a method which is able to capture changes of the high mountain cryosphere at very high spatio-temporal resolution (Ravanel et al., 2014). However, because the typical wavelengths emitted from former devices were absorbed by surfaces of fresh snow and bare ice, the application of TLS surveys in cryospheric sciences was first restricted to monitor the dynamics of rock walls (Rabatel et al., 2008; Abellán et al., 2009), rock glaciers (Bodin et al., 2008; Avian

et al., 2009; Kummert and Delaloye, 2015), or debris-covered glaciers (Conforti et al., 2005; Avian and Bauer, 2006). A few years ago, DEM creation of snowy and icy terrain using a new generation of terrestrial LiDAR devices operating in the near infrared range became possible (Prokop et al., 2008; Schwalbe et al., 2008; Grünewald et al., 2010; Egli et al., 2012). To our knowledge, Carturan et al. (2013) were the first to compute both reliable seasonal and annual geodetic mass balance of a very small glacier (Montasio Occidentale, 0.07 km$^2$, Julian Alps, Italy) from the differencing of repeated high-resolution TLS-

derived DEMs. Similar studies were to follow (e.g., López-Moreno et al., 2015, for the Monte Perdido Glacier in the Spanish



**Table 1.** Different parameters describing the characteristics of the five study sites listed from west to east: Location, surface area in 2010 ($A_{2010}$), relative area change between 1973 and 2010 ($\Delta A_{1973-2010}$), elevation range ($\Delta h_{2010}$) and length ($L_{2010}$) in 2010, and dominant aspect from the Swiss Glacier Inventory 2010 (SGI2010, Fischer et al., 2014). In addition, the dates of the field surveys are given.

| Parameters | Glacier de Prapio | Glacier du Sex Rouge | St. Annafirn | Schwarzbachfirn | Pizolgletscher |
|---|---|---|---|---|---|
| Location | 7.206°E 46.319°N | 7.214°E 46.327°N | 8.603°E 46.599°N | 8.612°E 46.597°N | 9.391°E 46.961°N |
| $A_{2010}$ (km$^2$) | 0.21 | 0.27 | 0.22 | 0.06 | 0.09 |
| $\Delta A_{1973-2010}$ (%) | −24.58 | −60.87 | −50.63 | −66.67 | −69.70 |
| $\Delta h_{2010}$ (m a.s.l.) | 2558–2854 | 2714–2867 | 2596–2928 | 2686–2832 | 2602–2783 |
| $L_{2010}$ (km) | 0.70 | 0.64 | 0.68 | 0.34 | 0.42 |
| Dominant aspect | NW | NW | N | NE | NE |
| **Dates of field surveys** | | | | | |
| summer 2013 | 14.09.* | 14.09. | 06.09. | 07.09. | 23.09. |
| summer 2014 | 22.09.* | 22.09. | 26.09. | 26.09. | 20.09. |
| summer 2015 | 21.09.* | 21.09. | 28.09. | 28.09. | 09.09. |

*only TLS surveys

Pyrenees). In the meantime, ground or Unmanned Aerial Vehicle (UAV) based close-range photogrammetry combined with computer vision algorithms such as Structure-from-Motion (SfM) has evolved into a valuable, cost-efficient and safe method to derive annual specific geodetic mass changes of small Alpine glaciers of similar quality compared to TLS or ALS techniques (Piermattei et al., 2015a, b). Although area-averaged geodetic mass balances calculated with high-resolution remote sensing

survey techniques showed close agreement with glacier-wide direct glaciological balances (Piermattei et al., 2015a), validation of these emerging new methods through comparison to in-situ measurements has so far been pending. It is, however, important to be conducted in order to know more about the quality and applicability of close-range high-resolution remote sensing techniques for glacier mass balance monitoring (Tolle et al., 2015).

In this paper we present a new data set of annual geodetic mass balances for five very small glaciers in the Swiss Alps

and two consecutive years (2013/14–2014/15) calculated from the differencing of repeated TLS-derived DEMs. The LiDAR surveys were performed with an ultra-long-range terrestrial laser scanner (*Riegl* VZ-6000) enabling the acquisition of surface elevation information over snow and ice of enhanced quality and over larger areas than with previous devices working in the near infrared (Deems et al., 2015; Gabbud et al., 2015). We compare our results to direct glaciological mass balances from very dense in-situ measurements and perform an in-depth uncertainty assessment of both the TLS-derived geodetic and the direct

glaciological mass changes. This finally allows us to comment on the potential of future mass balance monitoring of very small glaciers using our methodological approach.





**Figure 1.** Overview of study sites. (a) Glacier de Prapio and Glacier du Sex Rouge, (b) St. Annafirn and Schwarzbachfirn, and (c) Pizol-gletscher. The locations of the surveyed glaciers in the Swiss Alps is given in (d). Scan positions and horizontal view angles of the TLS surveys are shown by red dots and red dashed lines. Red numbers correspond to individual photographs of the study glaciers which were taken from the respective scan positions. Red triangles on the photographs refer to summits mentioned in the text. Black dashed areas indicate LiDAR shadow, and black crosses the locations of ablation stakes.



## 2 Study sites

The five study glaciers are located in the western, central, and eastern Swiss Alps (Fig. 1d). They are all smaller than $0.5\,\mathrm{km}^2$, generally north-exposed, and range from 2600 to 2900 m a.s.l. for the most part (Fig. 1, Tab. 1). In the context of a research project to better understand the response of very small glaciers in the Swiss Alps to climate change, the studied glaciers have
been subject to detailed scientific research since a couple of years, and a comprehensive set of empirical field data is now available for these previously unmeasured sites.

Glacier de Prapio is a steep and crevassed cirque glacier situated below the headwalls of a rock ledge which confines a nearby and flat valley glacier (Glacier de Tsanfleuron, see Hubbard et al., 2003, and references therein) to the west (Fig. 1a and 1-1). Observed area losses were comparatively moderate during past decades (Tab. 1).

Glacier du Sex Rouge lies west of the prominent Oldehore/Becca d'Audon (3123 m a.s.l.) (Fig. 1a, red triangle in Fig. 1-2). Over a flat ice divide, the glacier is connected to Glacier de Tsanfleuron (Fig. 1a). Since 1961, the glacier lost half of its volume (Fischer et al., 2015b). Apart from pronounced area and mass loss, the glacier surface flattened over the last decades. Today, no major crevasses exist at the glacier surface, meltwater runs off in meandering supraglacial or glacier marginal channels, and downglacier horizontal surface displacement rates measured with differential Global Positioning System (dGPS) amount
to $0.7\,\mathrm{m\,yr^{-1}}$ (M. Fischer, unpublished).

St. Annafirn is a cirque glacier protected by steep rock walls connecting St. Annahorn (2937 m a.s.l.) with Chastelhorn (2973 m a.s.l.) (Fig. 1b, red triangle in 1-3). Kenner et al. (2011) studied the erosion of these recently deglaciated rock walls using TLS, and Haberkorn et al. (2015) their thermal regime and its relation to snow cover. By 2010, St. Annafirn shrank to half its surface area in 1973, and lost about two thirds of its volume since 1986 (Tab. 1, Fischer et al. (2015b)).

Schwarzbachfirn is situated at the foot of the north face of Rothorn (2933 m a.s.l.) (Fig. 1b, red triangle in 1-4). Between 1990 and 2010, the glacier lost about 85% of its total volume (Fischer et al., 2015b). Since 1973, it retreated back to one third of its initial surface area (Tab. 1).

Pizolgletscher is located below the eastern headwalls of Pizol summit (2844 m a.s.l., red triangle in Fig. 1-5) and surrounded by rock walls on three sides (Fig. 1c and 1-5). According to first insights from direct glaciological observations of the
seasonal mass balance, Huss (2010) pointed to the remarkable small scale variability in accumulation and melt processes, to the importance of snow redistribution and the influence of albedo feedback mechanisms on the mass balance of very small glaciers. Pizolgletscher showed rapid retreat and lost 70% of its initial surface area between 1973 and 2010. Compared to 1961, a volume loss of 63% was observed (Tab. 1, Fischer et al. (2015b)).



## 3 Data and methods

### 3.1 TLS-derived surface elevation and mass changes

#### 3.1.1 The *Riegl* VZ-6000 terrestrial laser scanner

TLS is an active LiDAR technique, which measures the target distance based on the time-of-flight principle, i.e. the time of an
emitted laser signal required to return to its source (Deems et al., 2013). The ultra-long-range *Riegl* VZ-6000 terrestrial laser
scanner was applied here to produce annually repeated dense point clouds of the investigated glacier surfaces and adjacent areas,
which were subsequently used to create high-resolution DEMs. Using common TLS devices with wavelengths around 1500 nm,
i.e. with low reflectance but high absorption over snow and ice, the possible scanning distance is limited to a maximum of only
~150 m (Deems et al., 2013). These systems are therefore rather unsuitable for applications related to glacier monitoring.
Compared to the former generation of TLS systems operating also at the lower range of the near infrared but with relatively
shorter wavelengths (so-called Class 1 laser scanners, e.g., Carturan et al., 2013), the *Riegl* VZ-6000 Class 3B laser scanning
system allows faster surveys (up to 222'000 measurements s$^{-1}$) of larger areas and targets at greater distances (up to >6000 m)
with previously unprecedented accuracy and precision. Operating at 1064 nm, the TLS system is particularly well suited for
measuring snow- and ice-covered terrain (Tab. 2; RIEGL Laser Measurement Systems, 2013).

#### 15  3.1.2 LiDAR surveys

Starting in September 2013, TLS surveys of the study glaciers were performed on an annual basis, coinciding with in-situ
measurements on the same days to determine the direct glaciological mass balances (Tab. 1). Our approach to orient and
tie subsequent scans of the same study site into a common coordinate system was to use the relative orientation of the scans
and to define one scan as the reference to which the successive was registered (cf. section 3.1.3 below). Therefore, and also
because nearly the entire glacier surface areas were visible from selected scan locations, it was adequate to perform the LiDAR
surveys from only one single scan position for every site (red dots in Fig. 1). Additional measurements of Ground Control Points
(GCPs) with dGPS were omitted on purpose, as the potential of the ultra-long-range LiDAR system to remotely monitor surface
elevation and mass changes of very small glaciers was also to be tested with the aim of reducing laborious and potentially
dangerous in-situ measurements to a minimum here.
Prior to the laser scanning, the *Riegl* VZ-6000 was mounted on a tripod placed on stable surface in order to prevent ground
motion and reduced data acquisition quality (Fig. 2). Because Class 3B laser scanners operate at wavelengths not inherently
eye-safe, important precautions needed to be taken, including safety measures for the operators as well as people potentially
moving within a predefined ocular hazard distance (RIEGL Laser Measurement Systems, 2012).
    The scan parameters were chosen as a compromise between maximizing resolution (point density) over the measured area
and minimizing data acquisition time (Tab. 2, Supplementary Tab. 1). In order to avoid range ambiguity (several laser pulses
simultaneously in the air), the pulse repetition frequency was always set to 30 kHz. From the experience of the first TLS surveys
in autumn 2013, it was shown to be safe enough to increase the vertical and horizontal angle increments, i.e. the measurement



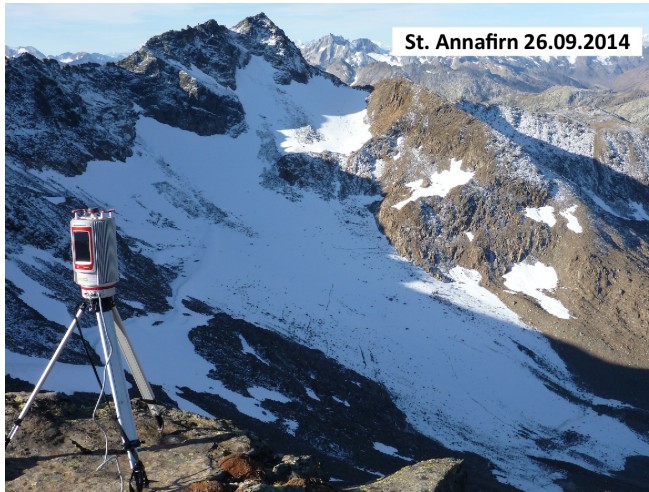

**Figure 2.** TLS survey of St. Annafirn in September 2014 with the *Riegl* VZ-6000 terrestrial laser scanner.

time, by one order of magnitude. This enhanced the ground resolution of target reflections (point density) to an important extent (Supplementary Tab. 1). The vertical angle range was often fully utilized, while the horizontal angle range (red dashed lines in Fig. 1a-c) had to be wider than the area of interest, i.e. the horizontal glacier extents, in order to use reflections from stable terrain outside the glaciers for the point cloud registration.

### 3.1.3 TLS data processing

Relative registration of the TLS point clouds was performed using RiSCAN PRO® v 2.1 (RIEGL Laser Measurement Systems, 2015). First, reflections outside the area of interest as well as clear outliers originating from atmospheric reflections due to moisture or dust were selected and permanently deleted. One point cloud of two consecutive scans was treated as registered. Changing surfaces (mostly reflections from snow and ice in our case) of the second point cloud were selected and temporarily removed until it only consisted of stable terrain, i.e. preferably planar and snow- and ice-free features as rock walls or rock outcrops and large boulders outside the glacier. Manual coarse registration was performed in order to approximatively shift the second point cloud into the local coordinate system of the registered one. Therefore, a sufficient number of spatially matching points (at least four) was identified by eye on both point clouds over stable terrain. Finally, a Multi-Station Adjustment (MSA) algorithm for semi-automatic fine registration using iterative closest point (ICP) techniques was performed (e.g., Zhang, 1992; Carrivick et al., 2012, cf. section 4.1 below). An octree filter was applied to the registered scans to remove noise and generate point clouds with equal numbers of reflections per area.

### 3.1.4 Calculation of surface elevation and geodetic mass changes

Interpolation of the processed point data sets to regular grids and calculation of surface elevation and geodetic mass changes was performed in ArcGIS. For each study site and individual time step, surface elevation changes $\Delta h_{\mathrm{TLS}}$ were calculated by





**Table 2.** Parameters and values of the *Riegl* VZ-6000 terrestrial laser scanner used for this study, and typical survey parameters chosen for the individual annual field surveys.

| System parameters | Value (range) |
|---|---|
| Measuring range for: | |
| - good diffusely reflective targets | >6000 m |
| - bad diffusely reflective targets | >2000 m |
| Ranging accuracy, precision | $\pm 15$ mm, $\pm 10$ mm |
| Measuring point frequency | 23'000–222'000 s$^{-1}$ |
| Measuring beam divergence | 0.12 mrad |
| Laser wavelength | Near infrared (1064 nm) |
| Scanning range: | |
| - horizontal | 0 to 360° |
| - vertical | –30 to 30° |
| Power supply | 11-28V DC |
| Temperature range: | |
| - operational | 0 to 40°C |
| - storage | –10 to 50°C |
| Weight | approx. 14.5 kg |
| Dimensions | 236 x 226.5 x 450 mm |
| **Chosen survey parameters** | **Parameter range** |
| Pulse repetition frequency | 30 kHz |
| Vertical angle increment | 0.08°(2013)/0.008°(2014,2015) |
| Vertical angle range | 60–120°from zenith |
| Horizontal angle increment | 0.08°(2013)/0.008°(2014,2015) |
| Horizontal angle range | 0–120° |

differencing of the TLS-derived DEMs. No extrapolations of $\Delta h_{\mathrm{TLS}}$ were performed for areas with LiDAR shadow as it would have introduced unnecessary uncertainty. This did not hamper a direct comparison between glaciological and geodetic mass balances because the relative proportions of LiDAR shadow over the glacier surface areas were minor (0.65% for Glacier du Sex Rouge, 7.87% for Glacier de Prapio, 3.33% for St. Annafirn, 1.18% for Schwarzbachfirn, and 15.76% for Pizolgletscher, Fig. 1). Glacier volume changes $\Delta V$ (m$^3$) for individual sites and years were derived by multiplying the area of the TLS-derived DEM of Difference (DoD) $A$ (m$^2$) with the mean surface elevation changes of all individual grid cells $\overline{\Delta h_{\mathrm{TLS}}}$.

Three basic approaches exist to convert geodetic volume to mass changes (e.g., Huss, 2013): (1) Application of a density of volume change of 900 kg m$^{-3}$ based on Sorge's law (Bader, 1954). This implies that neither changes in the mean firn density nor in the firn thickness and extent occur over time; (2) estimation of an average density of volume change to 850 kg m$^{-3}$ based





on typical changes in firn and ice volume over time (e.g., Zemp et al., 2013; Huss, 2013); (3) use of zonally variable conversion factors for firn and bare ice surfaces (e.g., Schiefer et al., 2007; Kääb et al., 2012). Based on the numerous field surveys (Supplementary Tab. 2) and limited ice dynamics, this third approach was applied here. Areas of bare ice, annual or multi-annual firn were manually defined for each glacier and survey date by considering repeated aerial and oblique photographs as well as direct observations. Corresponding densities of $900 \, \mathrm{kg \, m^{-3}}$ for ice $\rho_{\mathrm{ice}}$, $550 \, \mathrm{kg \, m^{-3}}$ for annual firn $\rho_{\mathrm{af}}$, and $700 \, \mathrm{kg}$ $\mathrm{m^{-3}}$ for multi-annual firn $\rho_{\mathrm{mf}}$ applied to calculate a glacier-wide volume-to-mass change conversion factor $f_{\Delta V}$ as

$$f_{\Delta V} = \frac{\Delta V_{\mathrm{ice}} \cdot \rho_{\mathrm{ice}}}{\Delta V} + \frac{\Delta V_{\mathrm{af}} \cdot \rho_{\mathrm{af}}}{\Delta V} + \frac{\Delta V_{\mathrm{mf}} \cdot \rho_{\mathrm{mf}}}{\Delta V}, \tag{1}$$

where $\Delta V_{\mathrm{ice}}$, $\Delta V_{\mathrm{af}}$ and $\Delta V_{\mathrm{mf}}$ correspond to the measured volume changes over areas of bare ice, annual or multi-annual firn. The TLS-derived specific geodetic mass balances $B_{\mathrm{TLS}}$ (m w.e. $\mathrm{yr^{-1}}$) were then derived by

$$B_{\mathrm{TLS}} = \frac{\Delta V \cdot f_{\Delta V}}{A \cdot \rho_{\mathrm{w}}}, \tag{2}$$

where $\rho_{\mathrm{w}}$ is the density of water. Neither firn compaction nor ice dynamics were considered to estimate the $f_{\Delta V}$ values. Due to the field observations, the spatio-temporal evolution of the firn thicknesses and extents during and prior to the measured years 2013–2015 could be assessed, and firn compaction assumed to be negligible as a result. Ice dynamics were likely negligible for the study glaciers as measured surface displacement rates were smaller than the resolution of the LiDAR-DEMs and dynamic thickening and thinning apparently smaller than the uncertainty in differences between TLS-derived and in-situ measured elevation changes at ablation stakes (cf. section 5.2 below).

If there was a significant amount of fresh snow at the time of the annual LiDAR surveys, additional snow depth measurements had to be carried out. Late summer snowfall events were recorded just a few days prior to the LiDAR surveys for Glacier de Prapio, Glacier du Sex Rouge (average of 0.18 m of fresh snow) and Pizolgletscher (0.20 m) in 2013 as well as for St. Annafirn and Schwarzbachfirn (both 0.30 m) in 2015. Snow probings on the glacier with a sufficient spatial coverage and density were performed on the same day as the LiDAR surveys, and recorded snow depth values inter- and extrapolated to the entire glacier surface. No snow depth measurements exist for Glacier de Prapio in 2013 due to difficult field site access. Average snow depth measured for the adjacent Glacier du Sex Rouge was therefore adopted. The final snow distribution grids were subtracted from the TLS-derived DoDs in order to calculate the actual annual volume and geodetic mass changes corrected for fresh snow.

## 3.2 Glaciological mass balance

### 3.2.1 In-situ measurements

Direct glaciological mass balance monitoring programmes including seasonal field observations started in 2006 (Pizolgletscher), 2012 (Glacier du Sex Rouge, St. Annafirn), and 2013 (Schwarzbachfirn) (Supplementary Tab. 2). Mostly in April, the spatial distribution and snow-water equivalent of the end-of winter snowpack accumulated on the glacier was determined through very





dense networks of manual snow probings and density measurements from snow pits (Supplementary Fig. 1, Supplementary Tab. 2). To quantify the amount of estival mass loss through snow and ice melt and to determine the annual mass balance, 4 m ablation stakes were drilled into the glaciers with a steam drill (see Fig. 1 for locations of individual stakes), and read out at the end of the melting season (September). Where mass gain through firn and snow accumulation occurred, snow pits were

dug during the summer surveys, and the accumulated volume was converted into mass applying typical values for snow and firn densities from the literature.

### 3.2.2 Derivation of the spatial mass balance distribution from in-situ measurements

A distributed accumulation and enhanced temperature-index melt model (Hock, 1999) was applied to derive the spatial surface mass balance distribution of the four measured glaciers in daily resolution. Using a semi-automated procedure, the model was

calibrated for each glacier and year individually to optimally match all seasonal field data. Taking into account the principal factors governing the accumulation and melt processes, the model can thus be regarded as a statistical tool for the spatio-temporal inter- and extrapolation of seasonal point mass balance measurements. In addition to the field measurements, required model inputs included updated glacier extents and DEMs of the glacier surfaces, as well as daily air temperature and precipitation data from nearby weather stations. A detailed description of the methodology is given in Huss et al. (2009) and/or Sold et al.

15 (2016).

## 4 Uncertainty assessment

### 4.1 TLS-derived geodetic mass changes

Uncertainty in the TLS-derived surface elevation changes presented here can be attributed to two main sources: (1) LiDAR data acquisition errors, and (2) data processing errors and DEM creation. In the following, important aspects of these errors

relevant to our methodological approach are assessed first. Then, we describe how we quantify the effective uncertainty in the TLS-derived surface elevation and geodetic mass changes.

Provided that the *Riegl* VZ-6000 used here operated reliably and ground motion was prohibited while scanning, errors in the acquired LiDAR point clouds of the surveyed glacier surfaces and surrounding areas are either terrain-induced or originate from the TLS system itself. Even though they vary in reality, for instance with the distance to the target, manufacturers commonly

provide simplified and constant values for the ranging accuracy and precision of their TLS systems. For the purpose of this study, the respective values given in Table 2 for the *Riegl* VZ-6000 are assumed to apply. Terrain-induced or geometric errors arise from the surface characteristics and orientation of the target relative to the scanner, i.e. slope and aspect, and from the laser-beam divergence, i.e. size of the laser-beam footprint at the target (Schürch et al., 2011; Deems et al., 2013; Hartzell et al., 2015).

Fine registration of two consecutive point clouds is an important LiDAR post-processing step and primarily enhances the quality of TLS-derived surface elevation changes (Prokop and Panholzer, 2009). For our study, fine registration using MSA/ICP





**Table 3.** Limits of detection for the TLS-derived surface elevation changes defined by the standard deviations of error (m) from the point cloud residuals ($\sigma_{\mathrm{MSA}}$) and number of points used for the Multi-Station Adjustment fine registration of consecutive point clouds in RiSCAN PRO® ($n$) for both observation periods 2013/14 and 2014/15 and the surveyed Glacier de Prapio (PRA), Glacier du Sex Rouge (SER), St. Annafirn (STA), Schwarzbachfirn (SWZ) and Pizolgletscher (PZL). In addition, the mean ($\mu$), median ($\tilde{x}$), standard deviation ($\sigma$) and interquartile range (iqr) of elevation differences from the comparison of TLS-derived annual surface elevation changes over stable terrain (all in m) are given.

| hydrological year | $\sigma_{\mathrm{MSA}}$ | $n$ | $\mu$ | $\tilde{x}$ | $\sigma$ | iqr |
|---|---|---|---|---|---|---|
| **PRA** | | | | | | |
| 2013/14 | 0.07 | 2940 | –0.18 | –0.20 | 0.49 | –0.74 to –0.07 |
| 2014/15 | 0.14 | 62902 | –0.05 | –0.03 | 1.11 | –0.79 to 0.72 |
| **SER** | | | | | | |
| 2013/14 | 0.18 | 1628 | –0.02 | –0.12 | 0.35 | –0.39 to 0.34 |
| 2014/15 | 0.05 | 48270 | –0.04 | –0.05 | 0.62 | –0.51 to 0.48 |
| **STA** | | | | | | |
| 2013/14 | 0.07 | 2486 | –0.01 | 0.03 | 0.36 | –0.07 to 0.12 |
| 2014/15 | 0.09 | 20896 | 0.17 | 0.18 | 0.59 | –0.08 to 0.42 |
| **SWZ** | | | | | | |
| 2013/14 | 0.17 | 7320 | –0.01 | –0.01 | 0.37 | –0.17 to 0.19 |
| 2014/15 | 0.08 | 11567 | 0.22 | 0.20 | 0.34 | 0.01 to 0.24 |
| **PZL** | | | | | | |
| 2013/14 | 0.07 | 1931 | 0.01 | –0.04 | 0.73 | –0.21 to 0.11 |
| 2014/15 | 0.14 | 10766 | 0.00 | –0.12 | 1.05 | –0.30 to 0.15 |

algorithms is indispensable as no absolute registration via GCPs is performed but only selected areas of the point clouds over stable terrain are used for the relative orientation of the scans. RiSCAN PRO® delivers error statistics of the final MSA results, including a full set of fitted point residuals. The standard deviation of error from the point residuals ($\sigma_{\mathrm{MSA}}$) can be used to quantify the quality of the registration process and define limits of detection for the TLS-derived surface elevation changes

5 (Gabbud et al., 2015). The latter ranged from ±0.05 to ±0.18 m (±0.12 m on average) over stable terrain surrounding the five glaciers in individual years (Tab. 3). Reliable results require use of some $10^4$ points for the MSA/ICP fine registration techniques applied here, the mean error to be zero and random errors Gaussian and pairwise uncorrelated (*Riegl*, personal communication). The latter two conditions are always fulfilled, whereas, due to the lower resolution (lower point densities) of the 2013 scans (Supplementary Tab. 1) and hence the lower number of points $n$ used for the fine registration of consecutive

10 point clouds (Tab. 3), MSA results and accordingly also registration quality have to be handled with some reservation for the observation period 2013/14.




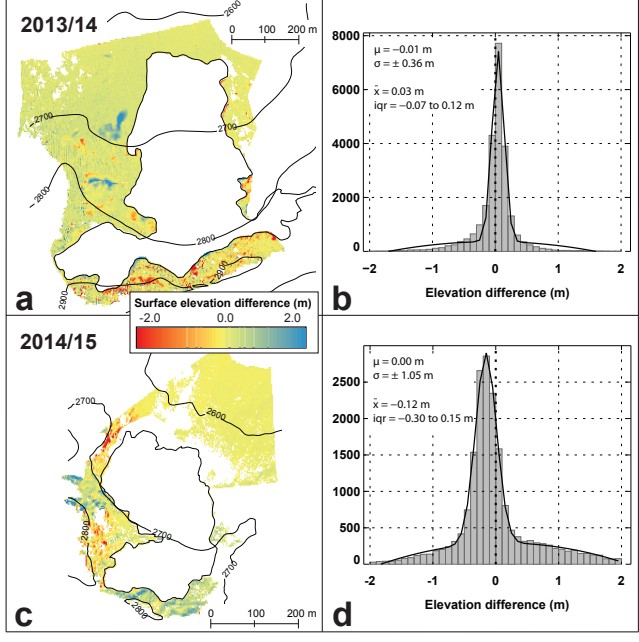

**Figure 3.** Comparison of annual surface elevation changes over stable terrain obtained by differencing of TLS-derived DEMs from two consecutive years. Spatial and corresponding frequency distributions of these changes for (a,b) St. Annafirn (2013/14) and (c,d) Pizolgletscher (2014/15). The black curves in (b) and (d) are normal fits over the data. In addition, the mean ($\mu$) and median ($\tilde{x}$) elevation differences, as well as corresponding standard deviations ($\sigma$) and interquartile ranges (iqr) are given.

Finally, point cloud filtering and gridding for DEM creation induces smoothing of the terrain information and additional error (Prokop and Panholzer, 2009; Deems et al., 2015). The quality of the TLS-derived DEMs is dependent on the LiDAR data acquisition and processing and the respective errors mentioned above. It increases with higher densities of points reflected from the surveyed terrain, which in turn depends on the chosen survey parameters (Tab. 2), surface characteristics, type and
flatness of the terrain (Carrivick et al., 2012).

The accuracy of the TLS-derived surface elevation changes and possible trends in elevation differences is assessed by comparison of consecutive DEMs over stable terrain outside the glaciers (Fig. 3). Except for Glacier de Prapio in 2013/14 (probably due to poor registration) as well as St. Annafirn and Schwarzbachfirn in 2014/15 (due to significant amounts of fresh snow in autumn 2015), the deviations from zero regarding the mean ($\mu$) and median ($\tilde{x}$) offsets over stable terrain between
consecutive DEMs are always smaller than the limits of detection ($\sigma_{\mathrm{MSA}}$) (Tab. 3, examples for St. Annafirn in 2013/14 and Pizolgletscher in 2014/15 in Fig. 3b,d). A significant trend towards higher biases in DEM differencing with steeper slopes is found (linear correlation coefficient $r = 0.96$). Remarkable elevation differences over comparably gently-sloping terrain can be attributed to changing surfaces such as snow patches (e.g., left of St. Annafirn, cf. Fig. 2 vs. 3a). For the steep rock walls confining St. Annafirn to the south, mean 2013/14 elevation differences are –0.23 m (standard deviation ±0.63 m)
(Fig. 3a). This corresponds to 3.5 times the mean annual erosion rates measured by Kenner et al. (2011) and hence points to





the limitations of TLS-derived elevation differences over very steep terrain (>50°) using our approach. Elevation differences outside Pizolgletscher for the period 2014/15 are also largest for the steep rock walls surrounding the glacier (Fig. 3c). Furthermore, probably some systematic error in one of the TLS-derived DEMs of Pizolgletscher results in a slight surface tilt (e.g., Lane et al., 2004) in lower left to upper right direction, which we, however, do not correct for.

The uncertainty in the average TLS-derived surface elevation changes ($\sigma_{\overline{\Delta h_{\mathrm{TLS}}}}$) for individual glaciers and years is assessed according to a simple implementation of Rolstad et al. (2009), and calculated with

$$\sigma_{\overline{\Delta h_{\mathrm{TLS}}}} = \pm\sqrt{\sigma^2_{\Delta h_{\mathrm{TLS}}} \cdot \frac{A_{\mathrm{cor}}}{5 \cdot A}}, \tag{3}$$

where $A_{\mathrm{cor}}$ is the range over which errors in DEM differencing are spatially correlated, conservatively estimated by $A_{\mathrm{cor}} = A$ here, and $\sigma_{\Delta h_{\mathrm{TLS}}}$ the standard deviation of error in TLS-derived glacier surface elevation changes, area-weighted for classes of equal surface slope, for which values are derived by taking the standard deviation of elevation differences per slope class from individual DoDs over stable terrain. Values for $\sigma_{\Delta h_{\mathrm{TLS}}}$ range from $\pm 0.26\,\mathrm{m}$ for the flatter glaciers like Glacier du Sex Rouge to $\pm 0.35\,\mathrm{m}$ for steeper glaciers like Glacier de Prapio.

The uncertainties in the densities of volume change for ice $\sigma_{\rho_{\mathrm{ice}}}$ (set to $\pm 20\,\mathrm{kg\,m^{-3}}$ here), annual and multi-annual firn $\sigma_{\rho_{\mathrm{af}}}$ and $\sigma_{\rho_{\mathrm{mf}}}$ (both assumed as $\pm 100\,\mathrm{kg\,m^{-3}}$) are used to estimate the uncertainty in the glacier-wide conversion factor $\sigma_{f_{\Delta V}}$ by

$$\sigma_{f_{\Delta V}} = \frac{\Delta V_{\mathrm{ice}} \cdot \sigma_{\rho_{\mathrm{ice}}}}{\Delta V} + \frac{\Delta V_{\mathrm{af}} \cdot \sigma_{\rho_{\mathrm{af}}}}{\Delta V} + \frac{\Delta V_{\mathrm{mf}} \cdot \sigma_{\rho_{\mathrm{mf}}}}{\Delta V}. \tag{4}$$

Finally, the uncertainty in the TLS-derived annual geodetic mass balance $\sigma_{B_{\mathrm{TLS}}}$ (m w.e.) is calculated following Huss et al. (2009) as

$$\sigma_{B_{\mathrm{TLS}}} = \pm\sqrt{(\overline{\Delta h_{\mathrm{TLS}}} \cdot \sigma_{f_{\Delta V}})^2 + (f_{\Delta V} \cdot \sigma_{\overline{\Delta h_{\mathrm{TLS}}}})^2 + \sigma_{\mathrm{s}}^2}, \tag{5}$$

where $\overline{\Delta h_{\mathrm{TLS}}}$ is the glacier-wide mean of TLS-derived surface elevation changes, and $\sigma_{\mathrm{s}}$ the uncertainty in correcting measured surface elevation changes for fresh snow, which is estimated by $\pm 20\%$ of the average measured snow depth. Resulting values for $\sigma_{B_{\mathrm{TLS}}}$ are listed in Table 4 and discussed in section 6 below.

## 4.2 Direct glaciological method

Uncertainty in both point and glacier-specific annual mass balance from direct field observations has often been estimated as $\pm 0.2\,\mathrm{m\ w.e.\ yr^{-1}}$ (e.g., Dyurgerov, 2002). In-depth assessments of random and systematic errors in glaciological mass balances of selected measured glaciers showed, however, that the actual uncertainty can significantly deviate from such static estimates (Thibert et al., 2008; Zemp et al., 2013; Beedle et al., 2014).

Uncertainty in the direct glaciological method either originates from measurement errors at individual point locations, from the representativeness of the measurement sites for their close surroundings, from the spatial inter- and extrapolation and





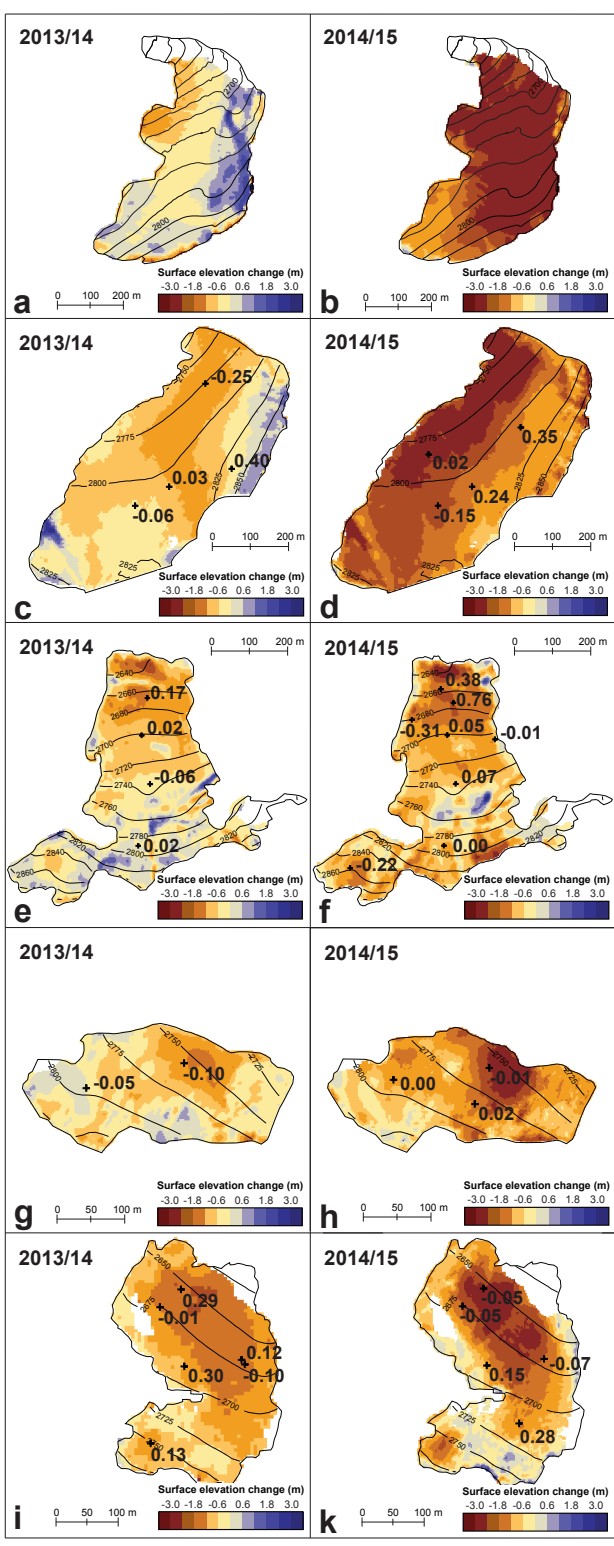

**Figure 4.** Spatial distribution of TLS-derived annual surface elevation changes $\Delta h_{\mathrm{TLS}}$ of (a,b) Glacier de Prapio, (c,d) Glacier du Sex Rouge, (e,f) St. Annafirn, (g,h) Schwarzbachfirn, and (i,k) Pizolgletscher in 2013/14 and 2014/15. The numbers indicate the differences between TLS-derived and in-situ measured annual elevation changes at ablation stakes ($\Delta h_{\mathrm{TLS}}$ minus $\Delta h_{\mathrm{direct}}$). White areas correspond to LiDAR shadow.





averaging of these results over the whole glacier, or from changes in the glacier geometry (Zemp et al., 2013). The latter is assumed to be negligible here as annually updated glacier outlines derived from high-resolution aerial imagery and in-situ GPS measurements are used. All stakes were located in the ablation areas of the glaciers in autumn 2014 and 2015. The uncertainty in the direct glaciological mass balance arising from measurement errors at individual stakes $\sigma_{\mathrm{abl}}$ (m w.e. yr$^{-1}$) is estimated

following Thibert et al. (2008) by $\sigma_{\mathrm{abl}} = 0.14/\sqrt{N_{\mathrm{abl}}}$, where $N_{\mathrm{abl}}$ is the number of ablation measurements for individual glaciers and years. Uncertainty from the spatial inter- and extrapolation of point measurements $\sigma_{\mathrm{int/ext}}$ (m w.e. yr$^{-1}$) can arise from a non-representative spatial distribution and/or insufficient density of stakes over the glacier, but is also related to settings of the method chosen for extrapolating the mass balance to the entire glacier. It is assessed by rerunning the mass balance model by Huss et al. (2009) used for calculating glacier-wide mass balance by closely constraining it with the seasonal

field data for each site and observation period with sets of melt parameters that differ from the calibrated ones by predefined ranges similar to those chosen by Kronenberg et al. (2016). In our approach, also the uncertainty in the measured winter snow accumulation distribution influences the spatial patterns of evaluated annual direct glaciological mass balance. It originates from both snow accumulation measurement errors $\sigma_{\mathrm{acc}}$ (m w.e. yr$^{-1}$) and errors in measured density of the winter snowpack $\sigma_{\rho_{\mathrm{s}}}$ (m w.e. yr$^{-1}$). The former is estimated by $\sigma_{\mathrm{acc}} = 0.21/\sqrt{N_{\mathrm{acc}}}$ (Thibert et al., 2008), where $N_{\mathrm{acc}}$ is the number of winter

snow depth measurements for individual glaciers and years, the latter by rerunning the mass balance model with values for $\sigma_{\rho_{\mathrm{s}}}$ that differ by $\pm 10\%$ from the measured ones. Finally, the uncertainty in the annual direct glaciological mass balance $\sigma_{B_{\mathrm{direct}}}$ (m w.e.) is calculated with

$$\sigma_{B_{\mathrm{direct}}} = \pm\sqrt{\sigma_{\mathrm{abl}}^2 + \sigma_{\mathrm{int/ext}}^2 + \sigma_{\mathrm{acc}}^2 + \sigma_{\rho_{\mathrm{s}}}^2}. \qquad (6)$$

Resulting values for $\sigma_{B_{\mathrm{direct}}}$ are listed in Table 4 and discussed in section 6 below.

## 5   Results

### 5.1   TLS-derived surface elevation and geodetic mass changes

Except for Glacier de Prapio in 2013/14 with balanced average conditions, all of the investigated very small glaciers showed clearly negative surface elevation and geodetic mass changes for the hydrological years 2013/14–2014/15 (Fig. 4, Tab. 4). In agreement with the different prevailing atmospheric conditions (especially in summer), measured mass losses were remarkably

stronger for the second time period (–1.65 m w.e. in 2014/15 averaged for the four glaciers measured with both methods compared to –0.59 m w.e. in 2013/14). Moreover, clear differences in glacier changes could be observed for individual sites in both years. In 2013/14, resulting surface elevation and mass changes of Pizolgletscher situated in the eastern Swiss Alps were significantly more negative compared to St. Annafirn and Schwarzbachfirn in the central Swiss Alps and Glacier de Prapio and Glacier du Sex Rouge in the western Swiss Alps. In 2014/15, though, the glaciers of the western Swiss Alps showed by far

the strongest mass losses. Measured changes were also more negative for glaciers of the central Swiss Alps, but only slightly more negative for Pizolgletscher compared to the precedent year. These regional differences in observed mass balances are





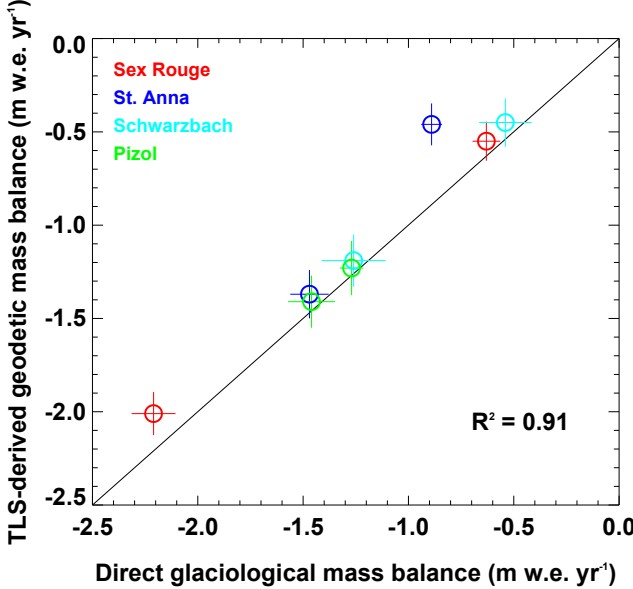

**Figure 5.** Direct glaciological vs. TLS-derived glacier-wide mass balances for Glacier du Sex Rouge, St. Annafirn, Schwarzbachfirn, and Pizolgletscher for the years 2013/14–2014/15. Error bars indicate the uncertainty ranges of both methods.

consistent with those reported for all monitored glaciers in Switzerland in 2013/14 and 2014/15 (Huss et al., 2015). The very small glaciers investigated here thus showed a similar response to the observed climatic forcing compared to larger glaciers, demonstrating that they are valuable indicators of mass balance variability despite their limited size.

A remarkable high small-scale variability in the spatial distribution of TLS-derived annual surface elevation changes is unveiled in Figure 4. Independent of the glacier-wide mean values varying for individual sites and years (Tab. 4), the variance in the TLS-derived surface elevation changes expressed by its $1\sigma$ standard deviation was similar for all glaciers ($\pm 0.81$ m on average evaluated over the small elevation ranges of about 100–300 m for individual glaciers). Moreover, a two-sample t-test showed that the magnitude of the spatial variability in $\Delta h_{\mathrm{TLS}}$ did not significantly change between 2013/14 and 2014/15, either.

## 5.2 Comparison to direct glaciological mass balances

The TLS-derived geodetic mass balances showed a close match with the direct glaciological mass balances extrapolated from in-situ measurements. Observed geodetic mass changes were slightly but systematically less negative compared to direct glaciological ones (Fig. 5, Tab. 4). Applying the statistical tests and approach proposed by Zemp et al. (2013), we validated the geodetic against the direct glaciological mass balances, and found that the differences in the resulting values between both methods were, except for St. Annafirn in 2013/14, not significant (95% confidence level). Corresponding uncertainties in the annual mass balances did, according to the results of a two-sample t-test, not differ significantly between both methods, either.



**Table 4.** Glacier-wide mean of TLS-derived surface elevation changes ($\overline{\Delta h_{\mathrm{TLS}}}$) (in m) as well as specific geodetic ($B_{\mathrm{TLS}}$) and direct glaciological ($B_{\mathrm{direct}}$) annual mass balance variables with corresponding uncertainties (in m w.e.) for Glacier de Prapio (PRA), Glacier du Sex Rouge (SER), St. Annafirn (STA), Schwarzbachfirn (SWZ) and Pizolgletscher (PZL) and the two observation periods 2013/14 and 2014/15.

| hydrological year | $\overline{\Delta h_{\mathrm{TLS}}}$ | $B_{\mathrm{TLS}}$ | $B_{\mathrm{direct}}$ |
| --- | --- | --- | --- |
| **PRA** | | | |
| 2013/14 | –0.01 | 0.02 ±0.53 | —— |
| 2014/15 | –3.19 | –2.58 ±0.15 | —— |
| **SER** | | | |
| 2013/14 | –0.66 | –0.55 ±0.10 | –0.63 ±0.07 |
| 2014/15 | –2.41 | –2.01 ±0.12 | –2.21 ±0.10 |
| **STA** | | | |
| 2013/14 | –0.49 | –0.46 ±0.11 | –0.89 ±0.05 |
| 2014/15 | –1.07 | –1.37 ±0.13 | –1.47 ±0.09 |
| **SWZ** | | | |
| 2013/14 | –0.55 | –0.45 ±0.13 | –0.54 ±0.12 |
| 2014/15 | –1.13 | –1.19 ±0.14 | –1.26 ±0.15 |
| **PZL** | | | |
| 2013/14 | –1.40 | –1.23 ±0.15 | –1.27 ±0.06 |
| 2014/15 | –1.40 | –1.41 ±0.14 | –1.46 ±0.11 |

The most prominent patterns in the observed spatial distribution of annual direct glaciological mass balances were captured by the TLS-derived surface elevation changes (Supplementary Fig. 1 vs. Fig. 4). The high-resolution LiDAR DEMs, however, allowed uncovering a level of detail in annual surface elevation changes unequalled by the direct mass balance observations.

In order to directly compare the extrapolated local mass balances from in-situ measurements to TLS-derived elevation changes, we multiplied the distributed direct glaciological mass balances with the reciprocal values of respective zonal conversion factors. Self-evidently, such confrontations of TLS-derived surface elevation changes to surface mass balances are only valid under the assumption that internal and basal mass balance components as well as dynamic thickening and thinning are negligible or absent. For larger glaciers, this is clearly not true (Fischer, 2011; Sold et al., 2013; Beedle et al., 2014). The comparison of differences between TLS-derived ($\Delta h_{\mathrm{TLS}}$) and in-situ measured ($\Delta h_{\mathrm{direct}}$) annual surface elevation changes at individual ablation stakes of Glacier du Sex Rouge, St. Annafirn, Schwarzbachfirn, and Pizolgletscher between 2013/14 and 2014/15 indicated, however, that dynamic thickening and thinning was minor and negligible for the very small glaciers studied here. The differences in surface elevation changes were close to zero for the majority of the point measurements (Fig. 6). Furthermore, the uncertainties in $\Delta h_{\mathrm{TLS}}$ minus $\Delta h_{\mathrm{direct}}$ calculated as $\pm\sqrt{\sigma^2_{\Delta h_{\mathrm{TLS}}} + \sigma^2_{\mathrm{abl}}}$ were mostly greater than the





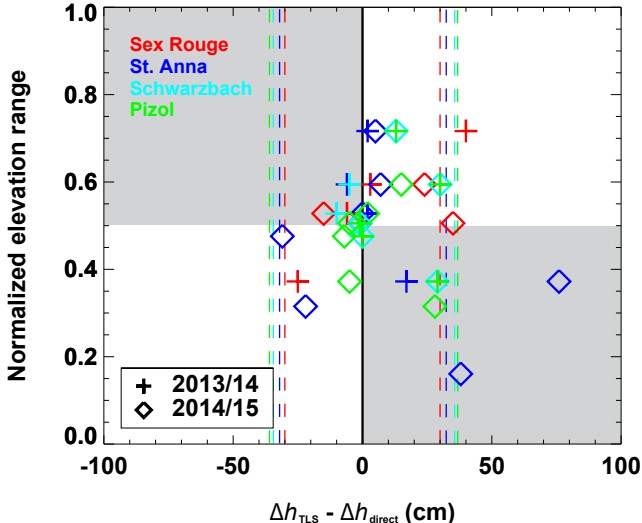

**Figure 6.** Differences between TLS-derived ($\Delta h_{\mathrm{TLS}}$) and observed ($\Delta h_{\mathrm{direct}}$) annual surface elevation changes at ablation stakes vs. normalized elevation range for Glacier du Sex Rouge, St. Annafirn, Schwarzbachfirn, and Pizolgletscher for the measured years 2013/14–2014/15. Dashed vertical lines indicate the uncertainty ranges of $\Delta h_{\mathrm{TLS}}$ minus $\Delta h_{\mathrm{direct}}$ relative to zero. Grey quadrants mark the theoretical ranges of $\Delta h_{\mathrm{TLS}}$ minus $\Delta h_{\mathrm{direct}}$ vs. normalized elevation range for glaciers in equilibrium.

measured differences themselves (Fig. 6), which would hamper robust statements about dynamic thickening and thinning. We therefore argue that applying reciprocal zonal conversion factors to the surface mass changes and hence also the spatial patterns of differences in $\Delta h_{\mathrm{TLS}}$ and $\Delta h_{\mathrm{direct}}$ as shown for Glacier du Sex Rouge and Pizolgletscher in Fig. 7 are presumably robust enough for qualitative but direct comparisons of both methods with respect to very small glaciers.

For these two glaciers and years, mean ($\mu$) and median ($\tilde{x}$) differences between $\Delta h_{\mathrm{direct}}$ and $\Delta h_{\mathrm{TLS}}$ were slightly negative on average, indicating that direct glaciological elevation changes were more negative than those based on the LiDAR DEMs (Fig. 7b,e). Over most of the glacier surface, both methods showed very similar results. On the other hand, remarkable differences in surface elevation changes were found for distinct, mostly steeper and/or glacier marginal areas (Fig. 7a,c,d,f). For Glacier du Sex Rouge in 2014/15, such disagreements were quite restricted to areas with no in-situ measurements. Furthermore, surface elevation changes over areas influenced by anthropogenic activity like a glacier walk maintained by snowcats (Fig. 7a) showed clear differences between $\Delta h_{\mathrm{TLS}}$ and $\Delta h_{\mathrm{direct}}$.

## 6  Applicability of the TLS system for mass balance monitoring of very small Alpine glaciers

On average, the uncertainty in the TLS-derived annual specific geodetic mass balances $\sigma_{B_{\mathrm{TLS}}}$ of the four very small glaciers in Switzerland measured with both methods is $\pm 0.13$ m w.e. yr$^{-1}$ (Tab. 4). The accuracy of our results is thus similar to geodetic





**Figure 7.** Difference between TLS-derived surface elevation changes and extrapolated glaciological mass balances which were converted to surface elevation changes by multiplication with the respective reciprocal values of zonal conversion factors. Spatial and corresponding frequency distributions as well as distributions of these changes vs. hypsometry for (a,b,c) Glacier du Sex Rouge (2014/15) and (d,e,f) Pizolgletscher (2013/14). The numbers in (a) and (d) indicate the differences between TLS-derived and in-situ measured annual elevation changes at ablation stakes ($\Delta h_{\mathrm{TLS}}$ minus $\Delta h_{\mathrm{direct}}$). The rectangle in (a) highlights a linear feature originating from a glacier walk maintained by snowcats. The black curves in (b) and (e) are normal fits over the data. In addition, the mean ($\mu$) and median ($\tilde{x}$) elevation differences, as well as corresponding standard deviations ($\sigma$) and interquartile ranges (iqr) are given.





mass changes computed over pentadal to decadal time periods based on high-resolution source data (e.g. Andreassen et al., 2015; Magnússon et al., 2016).

Uncertainty in the glacier-wide annual direct glaciological mass balances $\sigma_{B_{\mathrm{direct}}}$ of Glacier du Sex Rouge, St. Annafirn, Schwarzbachfirn and Pizolgletscher is $\pm 0.09$ m w.e. yr$^{-1}$ on average (Tab. 4), and hence comparable to the mean $\sigma_{B_{\mathrm{TLS}}}$. Resulting values for $\sigma_{B_{\mathrm{direct}}}$ presented here are similar to the one reported for Storglaciären (Jansson, 1999), but significantly smaller compared to the majority of all measured glaciers worldwide (e.g., Thibert et al., 2008; Beedle et al., 2014; Andreassen et al., 2015). This can be explained by the fact that the density of winter and summer point measurements is comparatively much more complete and substantially higher for our study glaciers than for most other glaciers (Supplementary Tab. 2; WGMS, 2013). Their very small surface area and the absence or minor fractions of very steep and/or heavily crevassed zones are, of course, optimal preconditions to measure direct glaciological mass balances with comparatively low uncertainty.

Hence, the quality of both the geodetic mass balances derived by repeated terrestrial LiDAR surveys and the direct glaciological mass balances extrapolated from very dense ins-situ measurements, which we use to validate our geodetic results, is very good. As resulting values of $B_{\mathrm{TLS}}$ for individual glaciers and years do not significantly differ from the respective values of $B_{\mathrm{direct}}$, we recommend the application of terrestrial laser scanning in general and our methodological approach in particular for future mass balance monitoring of very small Alpine glaciers. From our experience, however, a number of prerequisites need to be fulfilled in order to obtain reliable results, including (1) the absence of significant amounts of firn or fresh snow at the moment of the LiDAR survey at the end of the melting season; (2) the abundance and visibility of sufficient areas of stable terrain surrounding the entire glacier in order to achieve a good quality of the relative registration of consecutive LiDAR scans; and (3) good weather conditions (dry atmosphere). Frontal scan settings further increase the data quality.

As far as the first prerequisite mentioned above is concerned, the presence of significant amounts of fresh snow and/or firn on the glacier results in more error-prone conversions of TLS-derived volume to mass changes, even more if no additional in-situ measurements of their fraction and density are performed. On the other hand, from field evidence we know that along with the recorded atmospheric conditions (especially in summer) and the continuously negative mass balance context in the Swiss Alps over the last years (Supplementary Tab. 3), the studied very small glaciers hardly exhibit significant ratios of annual to perennial snow and firn anymore. This is of course in favour of reliable TLS-based geodetic mass balance monitoring applying our approach. Considerable firn extents and volumes can, however, rebuild on very small glaciers over very short time (e.g., within only one year) (Kuhn, 1995), which would again induce higher uncertainty in annual $B_{\mathrm{TLS}}$ values.

Following Ravanel et al. (2014), some further shortcomings of the ultra-long-range TLS system and our approach to derive annual surface elevation and geodetic mass changes of very small Alpine glaciers should not be ignored. Apart from the high costs for the purchase of the device itself and licenses for the data analysis software provided by the manufacturer, the complex and time-consuming post-processing of the LiDAR data should not be underestimated. The required level of expertise and experience with TLS data acquisition and processing is likely higher than for direct glaciological mass balance monitoring. In addition, the possibility to ensure safety, i. e. guarantee that no person other than the instrument operators wearing protection glasses move within a predefined ocular hazard distance, often proves to be non-trivial, especially in very well-developed areas like the Alps.





## 7 Conclusions

Despite their global predominance in absolute number, empirical field data on very small glaciers ($<0.5\,\mathrm{km}^2$) are currently sparse. In consequence, our understanding of their response to changes in the climatic forcing is still unsatisfactory. Monitoring surface elevation and mass changes of very small glaciers at high spatio-temporal resolution is a prerequisite to solve this

problem. Terrestrial laser scanning has evolved into a method which is able to fulfil these requirements. Because often almost the entire surface of very small glaciers is visible from one single location, it is a highly promising technique to create repeated high-resolution DEMs and subsequently compute geodetic surface elevation and mass changes of the smallest glaciers.

Here, we presented the application of an ultra-long-range terrestrial laser scanner (*Riegl* VZ-6000) especially designed for surveying snow- and ice-covered terrain. We derived annual surface elevation and geodetic mass changes of five very small

glaciers in Switzerland (Glacier de Prapio, Glacier du Sex Rouge, St. Annafirn, Schwarzbachfirn, and Pizolgletscher) over two consecutive years (2013/14–2014/15). Because validation of geodetic mass changes derived from repeated TLS surveys or other emerging close-range high-resolution remote sensing techniques is still pending, we carefully compared our results to direct glaciological mass balances from very dense in-situ measurements coinciding with the LiDAR surveys and performed an in-depth accuracy assessment of both methods.

Resulting surface elevation and geodetic mass changes were generally negative, with remarkably stronger mass losses for the second time period (–1.65 m w.e. in 2014/15 averaged for the four glaciers measured with both methods compared to –0.59 m w.e. in 2013/14), and apparently different regional mass balance patterns but stable and high small-scale variability in both years. TLS-derived specific geodetic mass balances were slightly less negative but did not vary significantly compared to direct glaciological mass balances extrapolated from in-situ measurements ($R^2 = 0.91$).

Uncertainty in the TLS-derived surface elevation changes can be attributed to LiDAR data acquisition errors, data processing errors and DEM creation. For geodetic mass changes, additional uncertainty results from the conversion of volume to mass changes. Mean uncertainty in the TLS-derived annual specific geodetic mass balances $\sigma_{B_{\mathrm{TLS}}}$ was $\pm0.13$ m w.e. Uncertainty in the direct glaciological mass balances was similar ($\pm0.09$ m w.e. $\mathrm{yr}^{-1}$ on average) and, due to the very dense in-situ measurements, rather small compared to the majority of measured glaciers worldwide.

Our results show that, under some restrictions (e.g., complex and time-consuming post-processing; absence of snow or significant amounts of firn at the time of the LiDAR surveys; abundance and visibility of sufficient areas of stable terrain surrounding the entire glacier; dry weather conditions), the TLS-based monitoring approach presented in this paper yields very accurate results and is therefore suitable for repeated mass balance measurements of very small Alpine glaciers. The most important shortcomings of our approach do not apply in a highly negative mass balance context, as observed for instance for

most of our field sites over precedent years. Under these circumstances, laborious, time-consuming, and potentially dangerous field measurements may be circumvented and the uncertain spatial inter- and extrapolation of point measurements over the whole glacier surface avoided.





*Acknowledgements.* This study is supported by the Swiss National Science Foundation (SNSF), grant 200021_137586. Sincere and many thanks to numerous friends and colleagues for their assistance in the field, and to Gstaad 3000 AG (K. von Siebenthal), Andermatt Gotthard Sportbahnen AG (C. Danioth), and Pizolbahnen AG for technical support and free transportation. Furthermore, we want to thank C. Gabbud, S. Gindraux, J.-B. Bosson and J. Carrivick for their earlier comments on this work.




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
