# Peer review of "Manuscript prepared for The Cryosphere with version 2016/01/22 8.09 Copernicus papers of the LATEX class copernicus.cls. Date: 16 May 2016"

_The Cryosphere, 2016_

## Referee Comment (RC1) · J. I. López-Moreno (Referee) · 5 Apr 2016

I have read the manuscript with great interest and my overall opinion is that the presented research is very interesting and the quality of the presentation is very good. Methodology is very well explained (even if needs some additional explanations) and the reliability of using very long range TLS for glaciological studies is clearly evidenced. I have a number of comments that authors may consider to use for preparing a revised version of the manuscript.

-Title: I wonder if the use of "ultra-long-range" is relatively standard, as it seems to me a bit "excessive". May be that using very long range is enough, or at least the text should inform that is a distance that has been very little used in previous research. As the

work deal more in validating the measurements rather than explaining the dynamics of the glacier, may be better to include some reference to the validation itself or the comparison to direct glaciological method (just a suggestion).

- Introduction. In page 2 line 4 when the importance of studying small glaciers is mentioned, it can be also stated that this is the very likely evolution of many current mid-size glaciers in areas like the Alps, and it is better to properly understand the dynamics of small glaciers when they are indeed very deteriorated.

- Study site: Page 5, line 9. "....were comparatively moderate during" The use of "moderate" is rather ambiguous, I would state that area losses were less than...or similar. In some part of the manuscript, probably here, a brief description of climate characteristics of the analyzed glaciers (and main differences if exits) and mentioning how was the climate during the two analyzed years compared to long-term climate (last decades) in the Swiss Alps would help to better interpret the presented data on mass balance.

- Data and methods: Page 6 line 31: -Which is the consequence of range ambiguity? a slightly expanded explanation (or a reference) might be useful.

- Page 7. Even if supplementary material inform of the characteristics of the point clouds, I would mention here some numbers about the most usual (or minimum) density of points acquired for this study. - What is an octree filter? -Page 9, line 5: Some reference to support the used densities for ice, annual and multi-annual firn? - Page 9, line 20, again, although this is presented as supplementary material some numbers on the density of snow depth measurements may be better than just saying "...with a sufficient spatial coverage..."

- Uncertainty assessment: Page 10 lines 22-24. Even if ground is stable, small instabilities may occur between the tips of the tripod and the bare rock, of if the ensemble of the tripod, or the tripod with the TLS is not properly ensured. - Where comes from that uncertainties of volume change for ice is set to +/- 20 kg m-3? - Page 15, line 9. I do not fully understand the procedure (rerunning the mass...) used here. - I think that authors

made their best to produce robust numbers on the uncertainty of the used methodology. They provide a very useful approach that may be replicated in future research. However, my feeling is the computation of each component of the uncertainty is based in equations and assumptions that are uncertain themselves. In this way, I think that in discussion (SECTION 6) .it should be remarked the difficulties to give exact numbers of uncertainty, that may vary spatially and also along the time, and at the end (at least in my opinion) an overall qualitative estimation that accumulated errors in the different methodological steps are much lower than observed changes in the elevation surface of the ice, is the most important, and it can be demonstrated when TLS estimations are directly compared with the changes in each ablation stake (Figure 5), or observing the annual changes in elevation surface over stable terrain (Figure 3).

References: It is cited a paper of our team that at the time of writing this paper was in TC discussion, and now is definitively published on TC., perhaps is better to change the citation: López-Moreno, J. I., Revuelto, J., Rico, I., Chueca-Cía, J., Julián, A., Serreta, A., Serrano, E., Vicente-Serrano, S. M., Azorin-Molina, C., Alonso-González, E., and García-Ruiz, J. M.: Thinning of the Monte Perdido Glacier in the Spanish Pyrenees since 1981, The Cryosphere, 10, 681-694, doi:10.5194/tc-10-681-2016, 2016.

Tables and figures: Table 1: i think it would be interesting to add the mean and maximum scanning distances for each glacier. Figure 1: Is it possible to provide pictures of the glaciers (1-5) just from the scanning positions? (It could ello saving Figure 2). Table 3. Probably there is space to write in the header what is each column, instead of using the symbols that needs a very long caption.

Hoping the comments will be useful,

Ignacio López

---

## Referee Comment (RC2) · Anonymous Referee #2 · 11 Apr 2016

The manuscript by Fischer et al. presents the application of a near infrared long-range terrestrial laser scanning to estimate the surface elevation changes and the mass balance using the geodetic method of five small glaciers in Switzerland. The geodetic mass balance changes were compared, for validation purpose, with the results obtained by a direct glaciological mass balance. The authors assessed the uncertainty and errors of both methods and demonstrated the feasibility of the remote sensing technique to estimate the mass balance. Moreover they highlighted the relevance of studying small glaciers as they can provide important insight into the atmospheric changes. The context of the research is well formulated in the introduction as they reported the main techniques currently used to calculate the mass balance, and they

highlighted the characteristics, advantages and gaps of the presented methods. The structure of the manuscript is correct, the methods are properly described and executed, and the results are interesting for the scientific community. In my opinion the paper deserves for the definitive publication in the Cryosphere. I include below very few comments/suggestions that could be taken in consideration in the final version of the manuscript. - Personally I consider more easy for reading to report the acronyms of the glaciers rather than the entire name. For example by including their acronym (Glacier de Prapio (PRA), Glacier du Sex Rouge (SER), St. Annafirn (STA), Schwarzbachfirn (SWZ) and Pizolgletscher (PZL), as you reported in the Table 3) in the Study sites section. If you change their name, then you should verify that you change it throughout the whole paper. - Page 7, line 9: I suggest to change "the second point cloud" with "the other point clouds" or "the unregistered point clouds" as one point cloud was fixed (e.g. the 2013 scan) and then the others two (e.g. the 2014 and 2015 point cloud), were co-registered using stable areas. Similar comment at line 12. - Page 11, line 5: Please change "the latter" with "$\sigma$MSA". - Page 12, line 7: (Fig. 3). Please add the name of the glaciers as done for Tab. 3, example..., line 10. (Fig. 3, examples for St. Annafirn and Pizolgletscher) - Page 18, line 14. As a first sentence of the Discussion section I suggest to make clear that the average value is for the glaciers measured with both TLS and ablation stakes. - Page 20, line 12: Ins-situ ->in-situ

---

## Referee Comment (RC3) · Anonymous Referee #3 · 26 Apr 2016

General comments ———————

This is a thorough assessment of the mass balance of five small glaciers in Switzerland using terrestrial laser scanning and direct glaciological measurements. The authors describe the methods, perform uncertainty assessments and give in general sufficient details on the study. In the title and throughout the term 'very small glaciers' is used. This term could be discussed in light of the literature. See also detailed comments for further comments and some suggestions for clarifications.

Detailed comments ———————

P1, L3: the sentence starting Since . . . does not read well.

P1, L4, are->have been. He sentence

P1, L9: and ->for

P1, L12: remove carefully

P1, L13: remove remarkably

P1, L15: remove very

P1, L22: remove always

P1, L21-22: on hence belong to the size class of very small glaciers (Huss, 2010).

text would flow better by defining small glaciers. Expalin what is used in the literature and explain your definition. In this paper we define small glaciers as …. (reference). Could refer to that there are different definitions, eg another paper in the cryosphere Bahr and Radic (2012) uses 1 km2 etc. The mass balance glossary does not define very small glaciers, but define Glacieret as 'A very small glacier, typically less than 0.25 km2 in extent, with no marked flow pattern visible at the surface' (Cogley et al, 2011) '

P2, L4-5: since most -> in)

P2, L19: This seems like a conclusion, but next sentence it is said that it is highly promising, rewrite.

P2, L25: Add e.g. before Zemp, as several authors have pointed this out, also earlier refs.

P3, L7: It is, however, …… -> Validation is needed to assess the quality … Note that in order is redundant, can replace throughout in the paper or at least in most Places

P3, L13: remove very

P3. L15: remove or reformulate last sentence.

P4, fig. 1: could add box around (d). I prefer a legend instead of having the explanation

of crosses and triangles etc. in the figure text. It is room for it in fig a above d. If text is kept, then Red numbers -> Numbers 1-5 Red triangles -> Triangles.

P5, L3-4: Just start with 'To better understand..' (remove words before)

P5, L5: since a couple of years is vague

P5, L6: delete 'for these previously unmeasured sites': later you talk about area and volume studies.

P5, L21. Retreated back to one third -> lost 2/3 of its area.

P5, L24. According to first insights: Rather state when measurements began.

P5, L27: glaciers. Mean in general or this glacier, clarify by writing 'of this glacier' if so.

P6, L21: But you do field work on some of these glaciers, would it not be interesting to compare wth dGPS measurements?

P7, L1: unclear what is meant by 'this' and 'to an important extent', be specific.

P7, L7: add commas after interest and after dust.

P7, L11: What does manual course registration mean? Could remove course?

P7, L19: could add reference as for RISCAN PRO, which edition was used?

P9, Table 2: What is the source, add manual reference.

P8, L4: Is three and four significant digits in the percentages justified? Would round it.

P8, L8-9: neither nor, -> assumes constancy of the density profile

P9, L1: could add that in the past other values have also been used, typically 900 kg/m3.

P9, L2: Based on the numerous . . ..here -> Based on information collected in field (supplementary Tab. 2), approach (3) was applied here.

P9, L12: Please add some more details on how the zones were mapped prior to the 2013 surveys. Doe this yield all 5 sites as one of them not measured and some began in 2012 and 2013 according to section 3.2.1?

P9, L14: Be specific on the displacement, e.g. <xx m/a

P9, L27: The information on the startup of the programmes could have been added in section 2.

P9, L28. Mostly -> Usually

P10, L14: Could add short details on the stations with reference to the data (e.g MeteoSwiss?) Use 'and' or 'or' not 'and/or'. Do you mean that you use their results or their methodology, a bit unclear.

Thes results would be interesting to compare with the spatial pattern found from the annual/biannual geodetic surveys, has this been compared?

P11, Table 3: Text. Divide sentence, is very long and hard to read.

P12, are not snow patches masked out?

P15, L22: can remove 'very small'

P15, L24.: Give results first, before interpreting them for better flow.

P15, L 25: why not give estimate for geodetic for allenfive?

P20, L7: This can be explained by the higher point density and more complete coverage than for most other glaciers (...)

P20, L10: Remove With comperatively low uncertainty.

P20, L11-L12. May remove the inserted clause, which we uses ..., for better flow.

P20, L14. Be specific on what you recommend here, specify your methodological approach.

P20, L20: Simplify by starting: Significant amounts of fresh snow or remaining firn on the glacier results is more error-prone . . . .

P20, L24: over the last years, which years?

P20. L25: Delete applying our approach.

P20, 28: Instead of 'Following' . . . Start sentence 'A disadvantage of using the TLS is the . . .' and refer to reference in parentheses.

P21, L2: . . .very small glaciers, here defined as (<0.5 km2) . . .

P21, L13,L23: very dense -> dense

P21, L28-32: Unclear sentences and ending, be specific.

References: ————-

Bahr, D. B. and Radić, V.: Significant contribution to total mass from very small glaciers, The Cryosphere, 6, 763-770, doi:10.5194/tc-6-763-2012, 2012.

Cogley, J.G., R. Hock, L.A. Rasmussen, A.A. Arendt, A. Bauder, R.J. Braithwaite, P. Jansson, G. Kaser, M. Möller, L. Nicholson and M. Zemp: Glossary of glacier mass balance and related terms, IHP-VII technical documents in hydrology No. 86, IACS Contribution No. 2., 2011.

---

## Author Comment (AC2) · 2 May 2016

- Personally I consider more easy for reading to report the acronyms of the glaciers rather than the entire name. For example by including their acronym (Glacier de Prapio (PRA), Glacier du Sex Rouge (SER), St. Annafirn (STA), Schwarzbachfirn (SWZ) and Pizolgletscher (PZL), as you reported in the Table 3) in the Study sites sec- tion. If you change their name, then you should verify that you change it throughout the whole paper.

We'd rather keep the glacier names as they are in the text and only have their acronyms in selected tables, as we did in the TCD version of our manuscript.

[Figure]

- Page 7, line 9: I suggest to change "the second point cloud" with "the other point clouds" or "the unregistered point clouds" as one point cloud was fixed (e.g. the 2013 scan) and then the others two (e.g. the 2014 and 2015 point cloud), were co-registered using stable areas. Similar comment at line 12

Now changed accordingly.

"Changing surfaces (mostly reflections from snow and ice in our case) of the second, unregistered point cloud were selected and temporarily removed until it only consisted of stable terrain, ..."

"Manual coarse registration was performed in order to approximatively shift the unregistered point cloud into the local coordinate system of the registered one."

- Page 11, line 5: Please change "the latter" with "$\sigma$MSA".

Done

"$\sigma\_MSA$ ranged from +/-0.05 to +/-0.18 m..."

- Page 12, line 7: (Fig. 3). Please add the name of the glaciers as done for Tab. 3, example..., line 10. (Fig. 3, examples for St. Annafirn and Pizolgletscher)

Implemented as suggested.

"The accuracy of the TLS-derived surface elevation changes and possible trends in elevation differences are assessed by comparison of consecutive DEMs over stable terrain outside the glaciers (examples for St. Annafirn in 2013/14 and Pizolgletscher in 2014/15 in Fig. 3)."

- Page 18, line 14. As a first sentence of the Discussion section I suggest to make clear that the average value is for the glaciers measured with both TLS and ablation stakes.

We think that this is already clear from our wording:

"the uncertainty in the TLS-derived annual specific geodetic mass balances $\sigma\_B\_TLS$

of the four very small glaciers in Switzerland measured with both methods is +/-0.13 m w.e. yr-1 (Tab. 4)."

- Page 20, line 12: Ins-situ ->in-situ

Done.

---

## Author Comment (AC3) · 2 May 2016

- Page 1, line 3: the sentence starting Since. . . does not read well.

Now rephrased.

"Investigating their mass balance, e.g. using the direct glaciological method, is a prerequisite to fill this knowledge gap. Since most recently, terrestrial laser scanning (TLS) techniques operating in the near infrared range have been successfully applied for the creation of repeated high-resolution digital elevation models and consecutive derivation of annual geodetic mass balances of very small glaciers."

- Page 1, line 4: are->have been. He sentence

Done.

- Page 1, line 9: and->for

We argue it's clearer not to change this (otherwise erroneous).

- Page 1, line 12: remove carefully

Done.

- Page 1, line 13: remove remarkably

Done.

- Page 1, line 15: remove very

Done.

- Page 1, line 22: remove always

Replaced with "so far" instead.

- Page 1, lines 21-22: on hence belong to the size class of very small glaciers (Huss, 2010). Text would flow better by defining small glaciers. Expalin what is used in the literature and explain your definition. In this paper we define small glaciers as... (reference). Could refer to that there are different definitions, eg another paper in the cryosphere Bahr and Radic (2012) uses 1 km2 etc. The mass balance glossary does not define very small glaciers, but define Glacieret as 'A very small glacier, typically less than 0.25 km2 in extent, with no marked flow pattern visible at the surface' (Cogley et al, 2011) '

Indeed, there are many different definitions of "very small glaciers" used if arbitrary area thresholds are used to classify individual glaciers as "very small glaciers". For upper thresholds, Bahr and Radic (2012) define 1.0 km2, Huss (2010) 0.5 km2, DeBeer and Sharp (2009) 0.4 km2, Cogley et al., (2011) 0.25 km2, Colucci and Guglielmin (2015) 0.1 km2.... Discussing these thresholds is beyond the scope here but will be included

in ongoing work which is not published yet. Nevertheless, we changed the wording in the abstract, introduction and conclusion in order to be clearer.

"Due to the relative lack of empirical field data, the response of very small glaciers (here defined as being smaller than 0.5 km2) to current atmospheric warming is not fully understood yet."

"Around 80% of the number of glaciers in the European Alps (Fischer et al., 2014; Gardent et al., 2014; Fischer et al., 2015; Smiraglia et al., 2015), and in mid- to low-latitude mountain ranges in general (Pfeffer et al., 2014), are smaller than 0.5 km2 and hence belong to the size class of very small glaciers according to the definition by Huss (2010)."

"Despite their global predominance in absolute number, empirical field data on very small glaciers, here defined as being smaller than 0.5 km2, are currently sparse."

- Page 2, lines 4-5: since most->in

Done.

- Page 2, line 19: This seems like a conclusion, but next sentence it is said that it is highly promising, rewrite.

Now slightly rephrased.

"Even though the initial costs of the scanner and software license are high, terrestrial laser scanning (TLS) techniques are generally easier and more cost-efficiently applied to individual sites and on the annual to seasonal timescale compared to ALS techniques (Heritage and Large, 2009). As often nearly the entire surface of very small glaciers is visible from on single location (e.g., from a frontal moraine, an accessible mountain crest or summit, or from the opposite valley side), TLS is particularly appropriate to generate high-resolution DEMs, as well as to derive annual geodetic mass balances of very small glaciers. Thus, laborious and time-consuming in-situ measurements could be circumvented, and the spatial inter- and extrapolation of point measure-

ments over the entire glacier surface avoided, which is known as an important source of uncertainty in direct glaciological mass balances (e.g., Zemp et al., 2015)."

- Page 2, line 25: Add e.g. before Zemp, as several authors have pointed this out, also earlier refs.

Done.

- Page 3, line 7: It is, however, . . . . . . -> Validation is needed to assess the quality. . .. Note that in order is redundant, can replace throughout the paper or at least in most places

Now implemented accordingly.

". . .validation of these emerging new methods through comparison to in-situ measurements has so far been pending. It is, however, needed to assess the quality and applicability of close-range high-resolution remote sensing techniques for glacier mass balance monitoring (Tolle et al., 2015)."

- Page 3, line 13: remove very

Done.

- Page 3, line 15: remove or reformulate last sentence

Now removed.

- Page 4, fig. 1: could add box around (d). I prefer a legend instead of having the explanation of crosses and triangles etc. in the figure text. It is room for it in fig a above d. If text is kept, then Red numbers->Numbers 1-5 Red triangles->Triangles.

We now added a box around (d). We'd rather keep the text instead of introducing a legend as we think it's clearer this way.

- Page 5, lines 3-4: Just start with 'To better understand. . .' (remove words before)

Done.

- Page 5, line 5: since a couple of years is vague

Now clarified.

"...the studied glaciers have been subject to detailed scientific research since 2006 (Pizolgletscher), 2012 (Glacier du Sex Rouge, St. Annafirn), and 2013 (Glacier de Prapio, Schwarzbachfirn), and a comprehensive set of empirical field data is now available for these sites."

- Page 5, line 6: delete 'for these previously unmeasured sites': later you talk about area and volume studies

Done.

- Page 5, line 21: Retreated back to one third->lost 2/3 of its area

We did not want to always write the same formulations for area changes of the individual study glaciers. The wording reviewer #3 suggests here is already used for other sites. Therefore, we'd rather keep this as it is.

- Page 5, line 24: According to first insights: Rather state when measurements began.

This becomes clear in section 3.2.1 below. – In addition, the first subset ("According to first insights from...") is now removed.

- Page 5, line 27: glaciers. Mean in general or this glacier, clarify by writing 'of this glacier' if so.

Now clarified.

"Huss (2010) pointed to the remarkable small scale variability in accumulation and melt processes, to the importance of snow redistribution and the influence of albedo feedback mechanisms on the mass balance of this very small glacier."

- Page 6, line 21: But you do field work on some of these glaciers, would it not be interesting to compare with dGPS measurements?

Good comment, could be done in the future, but we unfortunately did not have the data basis needed to do this in our study. dGPS measurements only exist for one of the five studied glaciers (Glacier du Sex Rouge), and comparison to TLS data is unfortunately not possible as only horizontal ice surface velocity but not vertical surface elevation changes were measured with dGPS.

- Page 7, line 1: unclear what is meant by 'this' and 'to an important extent', be specific

We erroneously wrote "increase" the vertical and horizontal angle increments instead of "decrease". Now changed accordingly. 'This' refers just to the precedent sentence, i.e. to "decrease the vertical and horizontal angle increments, i.e. increase the measurement time, by one order of magnitude". 'to an important exten' refers to the respective values in Supplementary Tab. 1. To clarify, we added a "cf." there (→"(cf. Supplementary Tab. 1)").

- Page 7, lines 6-7: add commas after interest and after dust

Done.

- Page 7, line 11: What does manual course registration mean? Could remove course?

It's "coarse" registration and not "course", "manual coarse registration" is a standard procedure in TLS data processing (see e.g. Deems et al., 2015, Cold Regions Science and Technology).

- Page 7, line 19: could add reference as for RISCAN PRO, which edition was used?

We argue that no reference is needed here, but we now refer to the edition (10.1) used.

- Page 8, Table 2: What is the source, add manual reference.

"RIEGL Laser Measurement Systems: Preliminary Data Sheet, 07.05.2013; RIEGL VZ-6000 - 3D Ultra long range terrestrial laser scanner with online waveform processing, RIEGL Laser Measurement Systems, Horn, Austria, 2013." Now added accordingly.

- Page 8, line 4: Is three and four significant digits in the percentages justified? Would round it.

Now rounded to two and three significant digits.

- Page 8, lines 8-9: neither nor, ->assumes constancy of the density profile

We argue that our initial formulation is clearer here.

- Page 9, line 1: could add that in the past other values have also been used, typically 900 kg/m3

This is already implemented with what we write just before "Three basic approaches exist to convert geodetic volume to mass changes (e.g., Huss, 2013): (1) Application of a density of volume change of 900 kg m-3 based on Sorge's law (Bader, 1954); (2)..."

- Page 9, line 2: Based on the numerous... here->Based on information collected in field (supplementary Tab. 2), approach (3) was applied here.

Implemented accordingly.

"Based on information collected during field surveys (Supplementary Tab. 2), approach (3) was applied here."

- Page 9, line 12: Please add some more details on how zones were mapped prior to the 2013 surveys. Does this yield all 5 sites as one of them not measured and some began in 2012 and 2013 according to section 3.2.1?

Done and yes.

"Due to field observations and repeated oblique and aerial orthoimagery, the spatio-temporal evolution of the firn thicknesses and extents during and prior to the measured years 2013–2015 could be assessed, and firn compaction assumed to be negligible as a result."

- Page 9, line 14: Be specific on the displacement, e.g. <xx m/a

[Figure]

Now implemented accordingly.

"Ice dynamics were likely negligible for the study glaciers as measured surface displacement rates were smaller than the resolution of the LiDAR DEMs (orders of magnitude 10-1 m yr-1 vs. 100 m) and..."

- Page 9, line 27: The information on the startup of the programmes could have been added in section 2.

Now we refer to section 3.2.1 in section 2 (see response above).

- Page 9, line 28: Mostly->Usually

Done.

- Page 10, line 14: Could add short details on the stations with reference to the data (e.g. MeteoSwiss?) Use 'and' or 'or' not 'and/or'. Do you mean that you use their results or their methodology, a bit unclear.

In our opinion, there is no need to give more details on the weather stations which refer to our meteo data used (beyond the scope here). All weather stations are indeed included in the MeteoSwiss network. We therefore now refer to MeteoSwiss in the text. 'and/or' is now replaced by 'or' only.

"... as well as daily air temperature and precipitation data from nearby MeteoSwiss weather stations. A detailed description of the methodology is given in Huss et al. (2009) or Sold et al. (2016)."

- These results would be interesting to compare with the spatial pattern found from the annual/biannual geodetic surveys, has this been compared?

Yes. Please refer to section 5.2 and Fig. 4 vs. Supplementary Fig. 1.

- Page 11, Table 3: Text. Divide sentence, is very long and hard to read.

Table caption is now shortened.

"Limits of detection for the TLS-derived surface elevation changes ($\sigma$_MSA) and number of points used for the Multi-Station Adjustment fine registration of consecutive point clouds (n) for both observation periods and the surveyed Glacier de Prapio (PRA), Glacier du Sex Rouge (SER), St. Annafirn (STA), Schwarzbachfirn (SWZ) and Pizolgletscher (PZL). In addition, the mean ($\mu$), median ($\sim$x), standard deviation ($\sigma$) and interquartile range (iqr) of elevation differences from the comparison of TLS-derived annual surface elevation changes over stable terrain (all in m) are given."

- Page 12, are not snow patches masked out?

Snow patches adjacent to the glaciers were, of course, not used for registration of two consecutive LiDAR point clouds. Nevertheless, they appear afterwards on the DEMs of Difference (as for instance on Fig. 3a which we refer to here).

- Page 15, line 22: can remove 'very small'

Done.

- Page 15, line 24: Give results first, before interpreting them for better flow

Implemented accordingly.

"Measured mass losses were remarkably stronger for the second time period (–1.65 m w.e. in 2014/15 averaged for the four glaciers measured with both methods compared to –0.59 m w.e. in 2013/14), which agrees well with the different prevailing atmospheric conditions (especially in summer) recorded during the observed years (MeteoSwiss, 2015, 2016)."

- Page 15, line 25: why not give estimate for geodetic for all five?

Because we could not validate the TLS-derived geodetic mass balance for Glacier de Prapio and explicitly refer to the glaciers "measured with both methods" here.

- Page 20, line 7: This can be explained by the higher point density and more complete coverage than for most other glaciers (. . .)

Implemented accordingly.

"This can be explained by the higher density and more complete coverage of winter and summer point measurements for our study glaciers than for most other glaciers (Supplementary Tab. 2; WGMS, 2013)."

- Page 20, line 10: Remove With comparatively low uncertainty

Done.

"Their very small surface area and the absence or minor fractions of very steep and/or heavily crevassed zones are, of course, optimal preconditions to accurately measure direct glaciological mass balances."

- Page 20, lines 11-12: May remove this inserted clause, which we use. . ., for better flow

Done.

"Hence, the quality of both the geodetic mass balances derived by repeated terrestrial LiDAR surveys and the direct glaciological mass balances extrapolated from dense in-situ measurements is very good."

- Page 20, line 14: Be specific on what you recommend here, specify your methodological approach

We now rephrased the sentence into a more general statement.

". . .we recommend the application of terrestrial laser scanning for future mass balance monitoring of very small Alpine glaciers."

- Page 20, line 20: Simplify by starting: Significant amounts of fresh snow or remaining firn on the glacier results is more error-prone. . ..

Done.

"Significant amounts of fresh snow or firn on the glacier results in more error-prone

conversions of TLS-derived volume to mass changes, even more if no additional in-situ measurements of their area fraction and density are performed.."

- Page 20, line 24: over the last years, which years?

Changed accordingly.

"On the other hand, from field evidence we know that along with the recorded atmospheric conditions (especially in summer) and the continuously negative mass balance context in the Swiss Alps over the last decade (Supplementary Tab. 3; Huss et al., 2015), the studied very small glaciers hardly exhibit significant ratios of annual to perennial snow and firn anymore."

- Page 20, line 25: Delete applying our approach

Done.

- Page 20, line 28: Instead of 'Following'. . . Start sentence 'A disadvantage of using the TLS is the. . ..' And refer to reference in parentheses.

Now implemented accordingly.

"Disadvantages of using the long-range TLS system and our approach to derive annual surface elevation and geodetic mass changes of very small Alpine glaciers are the high costs for the purchase of the device itself and licenses for the data analysis software provided by the manufacturer, as well as the complex and time-consuming post-processing of the LiDAR data. The required level of expertise and experience with TLS data acquisition and processing is likely higher than for direct glaciological mass balance monitoring (see e.g., Ravanel et al., 2014)."

- Page 21, line 2: . . .very small glaciers, here defined as (<0.5 km2). . .

Implemented accordingly.

"Despite their global predominance in absolute number, empirical field data on very

small glaciers, here defined as being smaller than 0.5 km2, are currently sparse."

- Page 21, lines 13,23: very dense -> dense

Done.

- Page 21, lines 28-32: Unclear sentences and ending, be specific

Implemented accordingly.

"Our results show that, under some restrictions, the TLS-based monitoring approach presented in this paper yields accurate results and is therefore suitable for repeated mass balance measurements of very small Alpine glaciers. The most important short-comings of our approach are related to the abundance of snow and firn at the time of the TLS surveys. They are insignificant in a highly negative mass balance context, as observed for instance for most of our field sites over precedent years. Under these circumstances, laborious, time-consuming, and potentially dangerous field measure-ments may be circumvented and the uncertain spatial inter- and extrapolation of point measurements over the whole glacier surface avoided."

---

## Author Response (AR1)

**Reviews of the paper tc-2016-46 submitted to The Cryosphere (Mauro Fischer, corresponding author):**

Dear Editor,

We want to thank for your work as the scientific editor of our paper.

We answered and commented on all points raised by the two reviewers below. – *Reviewer comments* are formatted in Times 12 italic, our response in Times 12 normal, and the corresponding revised text including information about the **corresponding line numbers in the new TC manuscript version** in Times 10 normal/bold.

**Comments by J. I. López-Moreno:**

*- Title: I wonder if the use of "ultra-long-range" is relatively standard, as it seems to me a bit "excessive". May be that using very long range is enough, or at least the text should inform that is a distance that has been very little used in previous research. As the work deal more in validating the measurements rather than explaining the dynamics of the glacier, may be better to include some reference to the validation itself or the comparison to direct glaciological method (just a suggestion)*

We now replaced "ultra-long-range" with "long-range" everywhere in the manuscript, including the title.

According to the reviewer's comments, we changed the title of our manuscript to "Application and validation of long-range terrestrial laser scanning to monitor the mass balance of very small glaciers in the Swiss Alps". To keep the title short, we did not directly include that validation is done against dense in-situ measurements/direct glaciological mass balances.

*- Introduction. In page 2 line 4 when the importance of studying small glaciers is mentioned, it can be also stated that this is the very likely evolution of many current mid-size glaciers in areas like the Alps, and it is better to properly understand the dynamics of small glaciers when they are indeed very deteriorated*

Now implemented accordingly with a new sentence.

**Page 1, Lns 58ff:**
"It is likely that currently medium-sized or even large glaciers become very small glaciers due to disintegration and substantial area loss over the next decades in areas like the European Alps (Zemp et al., 2006). A better understanding of their dynamics and sensitivity to climate change is thus important (Huss and Fischer, 2016)."

*- Study site: Page 5, line 9. "....were comparatively moderate during" The use of "moderate" is rather ambiguous, I would state that area losses were less than...or similar. In some part of the manuscript, probably here, a brief description of climate characteristics of the analyzed glaciers (and main differences if exits) and mentioning how was the climate during the two*

*analyzed years compared to long-term climate (last decades) in the Swiss Alps would help to better interpret the presented data on mass balance.*

We now implemented the first point as suggested

**Page 2, Lns 104f:**
"Observed area losses were smaller than for the other studied glaciers during past decades (Tab. 1)."

Concerning the reviewer's second point, we agree that data about the climate characteristics and variability of the study sites would help to better interpret the presented mass balance data (comparable to, for instance, López-Moreno et al., The Cryosphere 2016). However, we argue that such analyses go beyond the scope this study, which aims at validating TLS-derived annual geodetic mass balances of very small alpine glaciers with direct glaciological mass balances from dense in-situ measurements. We now refer to a new study by Huss and Fischer (2016, Frontiers in Eartch Science) in the revised version of the manuscript (Page 1, Ln 63) which is about the sensitivity of all very small glaciers in Switzerland to climate change. Further, we now also refer to the 2014 and 2015 annual climate bulletins of MeteoSwiss, which, if desired, will help the reader to better link the resulting measured mass balances as well as their regional and interannual variability to the prevailing atmospheric conditions during the observation periods 2013/14-2014/15 (cf. Page 10, Lns 24ff).

*- Data and methods: Page 6 line 31: -Which is the consequence of range ambiguity? A slightly expanded explanation (or a reference) might be useful.*

Now a slightly expanded explanation as well as a corresponding reference are given.

**Page 5, Lns 26ff:**
"In order to avoid range ambiguity and associated possible uncertainty due to several laser pulses simultaneously in the air (Rieger and Ullrich, 2012), the pulse repetition frequency was always set to 30 kHz."

*- Page 7. Even if supplementary material inform of the characteristics of the point clouds, I would mention here some numbers about the most usual (or minimum) density of points acquired for this study. - What is an octree filter?*

Now implemented accordingly.

**Page 5, Lns 33ff:**
"This enhanced the ground resolution of target reflections (point density) to an important extent. For all scans, average point density was 30 m$^{-2}$ (range 1 to 95 points m$^{-2}$, cf. Supplementary Tab. 1)."

An octree filter segments the point cloud into cubes of selected length x, width y, and height z, reducing the data within each cube to a single point. We now complemented this sentence with an exemplary reference.

**Page 6, Lns 5ff:**
"An octree filter (e.g., Perroy et al., 2010) was applied to the registered scans to remove noise and generate point clouds with equal numbers of reflections per area."

*-Page 9, line 5: Some reference to support the used densities for ice, annual and multi-annual firn?*

We argue that the density for ice chosen is standard. We added a reference supporting the chosen densities for annual and multi-annual firn.

**Page 6, Lns 42ff:**
"Corresponding densities of 900 kg m$^{-3}$ for ice $\varrho_{ice}$, 550 kg m$^{-3}$ for annual firn $\varrho_{af}$ , and 700 kg m$^{-3}$ for multi-annual firn $\varrho_{mf}$ (e.g., Sold et al., 2015) applied to calculate a glacier-wide volume-to-mass change conversion factor…"

*- Page9, line 20, again, although this is presented as supplementary material some numbers on the density of snow depth measurements may be better than just saying "...with a sufficient spatial coverage..."*

Here, we do not refer to winter snow accumulation measurements, but to snow probings performed if there was a significant amount of fresh snow at the time of the annual LiDAR survey in autumn (as mentioned in the text). – Autumnal fresh snow covers are, from our experience, usually spatially much more homogeneous than end-of-winter snow accumulation patterns, so a smaller density compared to the winter surveys was enough. We now refer to the recorded median density (as a number of measurements per square kilometer) of autumnal snow probings on the respective glaciers.

**Page 6, Lns 74ff:**
"Snow probings on the glaciers with a complete spatial coverage and a median density of about 200 measurements km$^{-2}$ were performed on the same days as the LiDAR surveys, and measured snow depth values inter- and extrapolated to the entire glacier surfaces."

*- Uncertainty assessment: Page 10 lines 22-24. Even if ground is stable, small instabilities may occur between the tips of the tripod and the bare rock, of if the ensemble of the tripod, or the tripod with the TLS is not properly ensured.*

We fully agree. – That's why we wrote "**Provided that** the RieglVZ-6000 used here operated reliably and ground motion was prohibited while scanning, …". So our formulation already implies that for instance small instabilities between the tips of the tripod and the bare rock may occur.

*- Where comes from that uncertainties of volume change for ice is set to +/- 20 kg m-3?*

This is just a conservative estimate for a range in ice density for small mountain glaciers. Now clarified.

**Page 8, Lns 55f:**
"…(estimated as +/- 20 kg m$^{-3}$ here)…"

*- Page 15, line 9. I do not fully understand the procedure (rerunning the mass...) used here. - I think that authors made their best to produce robust numbers on the uncertainty of the used methodology. They provide a very useful approach that may be replicated in future research. However, my feeling is the computation of each component of the uncertainty is based in equations and assumptions that are uncertain themselves. In this way, I think that in discussion (SECTION 6) .it should be remarked the difficulties to give exact numbers of uncertainty, that may vary spatially and also along the time, and at the end (at least in my opinion) an overall qualitative estimation that accumulated errors in the different methodological steps are much lower than observed changes in the elevation surface of the ice, is the most important, and it can be demonstrated when TLS estimations are directly compared with the changes in each ablation stake (Figure 5), or observing the annual changes in elevation surface over stable terrain (Figure 3).*

Now clarified.

**Page 9, Lns 44ff:**
"$\sigma\_int/ext$ is assessed by rerunning the mass balance model by Huss et al. (2009) used for calculating glacier-wide mass balance (cf. section 3.2.2) by closely constraining it with the seasonal field data for each site and observation period but melt parameters and temperature lapse rates that differed from the reference values by predefined ranges (cf. Kronenberg et al., 2016)."

We now remind the reader of this important issue in the discussion (section 6).

**Page 12, Lns 6ff:**
"Even though we consider our approach to quantify both $\sigma\_B\_TLS$ and $\sigma\_B\_direct$ as robust and promote its application to similar studies in the future, we want to remind the reader that it is generally difficult to give exact numbers of such uncertainties, and that each component of $\sigma\_B\_TLS$ and $\sigma\_B\_direct$ is based on assumptions that are, to some extent, uncertain themselves. By its nature, the stochastic uncertainty in the glacier-wide TLS-derived geodetic mass balance $\sigma\_B\_TLS$ is much lower than the potential error in the observed surface elevation changes for single pixels, as estimated for instance from the comparison of DoDs over stable terrain (Fig. 3)."

*References: It is cited a paper of our team that at the time of writing this paper was in TC discussion, and now is definitively published on TC., perhaps is better to change the citation: López-Moreno, J. I., Revuelto, J., Rico, I., Chueca-Cía, J., Julián, A., Serreta, A., Serrano, E., Vicente-Serrano, S. M., Azorin-Molina, C., Alonso-González, E., and García-Ruiz, J. M.: Thinning of the Monte Perdido Glacier in the Spanish Pyrenees since 1981, The Cryosphere, 10, 681-694, doi:10.5194/tc-10-681-2016, 2016.*

We now refer to the revised TC (2016) version of the corresponding study by López-Moreno et al. Thanks.

*Tables and figures: Table 1: I think it would be interesting to add the mean and maximum scanning distances for each glacier.*

We think that this would be a bit misplaced in Table 1. Nevertheless, the reader already has this information, as mean and maximum scanning distances related to the study glaciers' extents are already nicely visible on Figure 1.

*Figure 1: Is it possible to provide pictures of the glaciers (1-5) just from the scanning*

*positions? (It could ello saving Figure 2).*

Actually, all pictures (1-5) in Figure 1 were taken just from the respective scanning positions. This is also mentioned in the figure caption ("Red numbers correspond to individual photographs of the study glaciers which were taken from the respective scan positions"). We'd rather keep Figure 2 to show the scan setting and installation of the tripod/scanner on stable ground, as this is also referred to in the methods part of our manuscript.

*Table 3. Probably there is space to write in the header what is each column, instead of using the symbols that needs a very long caption*

No, there actually isn't, sorry. We are aware of the rather long table caption but argue that it remains clearer and better understandable for the reader to keep Table 3's caption and heading as in the TCD version of our manuscript.

**Comments by Anonymous Referee #2:**

*- Personally I consider more easy for reading to report the acronyms of the glaciers rather than the entire name. For example by including their acronym (Glacier de Prapio (PRA), Glacier du Sex Rouge (SER), St. Annafirn (STA), Schwarzbachfirn (SWZ) and Pizolgletscher (PZL), as you reported in the Table 3) in the Study sites section. If you change their name, then you should verify that you change it throughout the whole paper.*

We'd rather keep the glacier names as they are in the text and tables.

*- Page 7, line 9: I suggest to change "the second point cloud" with "the other point clouds" or "the unregistered point clouds" as one point cloud was fixed (e.g. the 2013 scan) and then the others two (e.g. the 2014 and 2015 point cloud), were co-registered using stable areas. Similar comment at line 12*

Now changed accordingly.

**Page 5, Lns 49ff:**
"Changing surfaces (mostly reflections from snow and ice in our case) of the second, unregistered point cloud were selected and temporarily removed until it only consisted of stable terrain, …"

**and Page 5, Lns 54ff:**
"Manual coarse-registration was performed in order to approximatively shift the unregistered point cloud into the local coordinate system of the registered one."

*- Page 11, line 5: Please change "the latter" with "σMSA".*

Done

**Page 7, Ln 61:**
"$\sigma_{MSA}$ ranged from +/-0.05 to +/-0.18 m…"

*- Page 12, line 7: (Fig. 3). Please add the name of the glaciers as done for Tab. 3, example…, line 10. (Fig. 3, examples for St. Annafirn and Pizolgletscher)*

Implemented as suggested.

**Page 8, Lns 13ff:**
"The accuracy of the TLS-derived surface elevation changes and possible trends in elevation differences are assessed by comparison of consecutive DEMs over stable terrain outside the glaciers (examples for St. Annafirn in 2013/14 and Pizolgletscher in 2014/15 in Fig. 3)."

*- Page 18, line 14. As a first sentence of the Discussion section I suggest to make clear that the average value is for the glaciers measured with both TLS and ablation stakes.*

We think that this is already clear from our wording:

**now Page 11, Lns 75ff:**
 "On average, the uncertainty in the TLS-derived annual specific geodetic mass balances $\sigma\_B\_{TLS}$ **of the four very small glaciers** in Switzerland **measured with both methods** is +/-0.13 m w.e. yr-1 (Tab. 4)."

*- Page 20, line 12: Ins-situ ->in-situ*

Done.

**Comments by Anonymous Referee #3:**

*- Page 1, line 3: the sentence starting Since… does not read well.*

Now rephrased.

**Page 1, Lns 4ff:**
"Investigating their mass balance, e.g., using the direct glaciological method, is a prerequisite to fill this knowledge gap. Terrestrial laser scanning (TLS) techniques operating in the near infrared range can be applied for the creation of repeated high-resolution digital elevation models and consecutive derivation of annual geodetic mass balances of very small glaciers."

 *- Page 1, line 4: are->have been. He sentence*

Done.

*- Page 1, line 9: and->for*

We argue it's clearer not to change this (otherwise erroneous).

*- Page 1, line 12: remove carefully*

Done.

*- Page 1, line 13: remove remarkably*

Done.

*- Page 1, line 15: remove very*

Done.

*- Page 1, line 22: remove always*

Replaced with "so far" instead.

*- Page 1, lines 21-22: on hence belong to the size class of very small glaciers (Huss, 2010). Text would flow better by defining small glaciers. Expalin what is used in the literature and explain your definition. In this paper we define small glaciers as... (reference). Could refer to that there are different definitions, eg another paper in the cryosphere Bahr and Radic (2012) uses 1 km2 etc. The mass balance glossary does not define very small glaciers, but define Glacieret as 'A very small glacier, typically less than 0.25 km2 in extent, with no marked flow pattern visible at the surface' (Cogley et al, 2011) '*

Indeed, there are many different definitions of "very small glaciers" used if arbitrary area thresholds are used to classify individual glaciers as "very small glaciers". For upper thresholds, Bahr and Radic (2012) define $1.0$ km$^2$, Huss (2010) $0.5$ km$^2$, DeBeer and Sharp (2009) $0.4$ km$^2$, Cogley et al., (2011) $0.25$ km$^2$, Colucci and Guglielmin (2015) $0.1$ km$^2$.... We have the impression that discussing these thresholds is beyond the scope of the present paper. Nevertheless, we changed the wording in the abstract, introduction and conclusions in order to be clearer.

**Page 1, Lns 1ff:**
"Due to the relative lack of empirical field data, the response of very small glaciers (here defined as being smaller than $0.5$ km$^2$) to current atmospheric warming is not fully understood yet."

**Page 1, Lns 37ff:**
"Around 80% of the *number* of glaciers in the European Alps (Fischer et al., 2014; Gardent et al., 2014; Fischer et al., 2015; Smiraglia et al., 2015), and in mid- to low-latitude mountain ranges in general (Pfeffer et al., 2014), are smaller than $0.5$ km$^2$ and hence belong to the size class of very small glaciers according to the definition by Huss (2010)."

**Page 13, Lns 48ff:**
"Despite their global predominance in absolute number, empirical field data on very small glaciers, here defined as being smaller than $0.5$ km$^2$, are currently sparse."

*- Page 2, lines 4-5: since most->in*

Done.

*- Page 2, line 19: This seems like a conclusion, but next sentence it is said that it is highly promising, rewrite.*

Now slightly rephrased.

**Page 2, Lns 23ff:**
"Even though the initial costs of the scanner and software license are high, terrestrial laser scanning (TLS) techniques are generally easier and more cost-efficiently applied to individual sites and on the annual to seasonal timescale compared to ALS techniques (Heritage and Large, 2009). As often nearly the entire surface of very small glaciers is visible from one single location (e.g., from a frontal moraine, an accessible mountain crest or summit, or from the opposite valley side), TLS is particularly appropriate to generate high-resolution DEMs, as well as to derive annual geodetic mass balances of very small glaciers. Thus, laborious and time-consuming in-situ measurements could be circumvented, and the spatial inter- and extrapolation of point measurements over the entire glacier surface avoided, which is known as an important source of uncertainty in direct glaciological mass balances (e.g., Zemp et al., 2013)."

*- Page 2, line 25: Add e.g. before Zemp, as several authors have pointed this out, also earlier refs.*

Done.

*- Page 3, line 7: It is, however, …… -> Validation is needed to assess the quality…. Note that in order is redundant, can replace throughout the paper or at least in most places*

Now implemented accordingly.

**Page 2, Lns 70ff:**
"…validation of these emerging new methods through comparison to in-situ measurements has so far been pending. It is, however, needed to assess the quality and applicability of close-range high-resolution remote sensing techniques for glacier mass balance monitoring (Tolle et al., 2015)."

*- Page 3, line 13: remove very*

Done.

*- Page 3, line 15: remove or reformulate last sentence*

Now removed.

*- Page 4, fig. 1: could add box around (d). I prefer a legend instead of having the explanation of crosses and triangles etc. in the figure text. It is room for it in fig a above d. If text is kept, then Red numbers->Numbers 1-5 Red triangles->Triangles.*

We now added a box around (d). We'd rather keep the text instead of introducing a legend as we think it's clearer this way.

*- Page 5, lines 3-4: Just start with 'To better understand...' (remove words before)*

Done.

*- Page 5, line 5: since a couple of years is vague*

Now clarified.

**Page 2, Lns 95ff:**
"…the studied glaciers have been subject to detailed scientific research since 2006 (Pizolgletscher), 2012 (Glacier du Sex Rouge, St. Annafirn), and 2013 (Schwarzbachfirn), and a comprehensive set of empirical field data is now available for these sites."

*- Page 5, line 6: delete 'for these previously unmeasured sites': later you talk about area and volume studies*

Done.

*- Page 5, line 21: Retreated back to one third->lost 2/3 of its area*

We did not want to always write the same formulations for area changes of the individual study glaciers. The wording reviewer #3 suggests here is already used for other sites. Therefore, we'd rather keep this as it is.

*- Page 5, line 24: According to first insights: Rather state when measurements began.*

This becomes clear in section 3.2.1 below. – In addition, the first subset ("According to first insights from…") is now removed.

*- Page 5, line 27: glaciers. Mean in general or this glacier, clarify by writing 'of this glacier' if so.*

Now clarified.

**Page 4, Lns 25ff:**
"Huss (2010) pointed out the remarkable small-scale variability in accumulation and melt processes, to the importance of snow redistribution and the influence of albedo feedback mechanisms on the mass balance of this very small glacier."

*- Page 6, line 21: But you do field work on some of these glaciers, would it not be interesting to compare with dGPS measurements?*

Good comment, could be done in the future, but we unfortunately did not have the data basis needed to do this in our study. dGPS measurements only exist for one of the five studied glaciers (Glacier du Sex Rouge), and comparison to TLS data is

unfortunately not possible as only horizontal ice surface velocity but not vertical surface elevation changes were measured with dGPS.

*- Page 7, line 1: unclear what is meant by 'this' and 'to an important extent', be specific*

We erroneously wrote "increase" the vertical and horizontal angle increments instead of "decrease". Now changed accordingly. 'This' refers just to the precedent sentence, i.e. to "decrease the vertical and horizontal angle increments, i.e. increase the measurement time, by one order of magnitude". 'to an important extent' refers to the respective values in Supplementary Tab. 1. To clarify, we added a "cf." there (→"(cf. Supplementary Tab. 1)").

*- Page 7, lines 6-7: add commas after interest and after dust*

Done.

*- Page 7, line 11: What does manual course registration mean? Could remove course?*

It's "coarse" registration and not "course", "manual coarse-registration" is a standard procedure in TLS data processing (see e.g. Deems et al., 2015, Cold Regions Science and Technology).

*- Page 7, line 19: could add reference as for RISCAN PRO, which edition was used?*

We did not found a comparable reference here, but we now refer to the edition (10.1) used.

*- Page 8, Table 2: What is the source, add manual reference.*

"RIEGL Laser Measurement Systems: Preliminary Data Sheet, 07.05.2013; RIEGL VZ-6000 - 3D Ultra long range terrestrial laser scanner with online waveform processing, RIEGL Laser Measurement Systems, Horn, Austria, 2013."
Now added accordingly.

*- Page 8, line 4: Is three and four significant digits in the percentages justified? Would round it.*

Now rounded to two and three significant digits.

*- Page 8, lines 8-9: neither nor, ->assumes constancy of the density profile*

We argue that our initial formulation is clearer here.

*- Page 9, line 1: could add that in the past other values have also been used, typically 900 kg/m3*

This is already implemented with what we write just before "Three basic approaches exist to convert geodetic volume to mass changes (e.g., Huss, 2013): (1) Application of a density of volume change of 900 kg m-3 based on Sorge's law (Bader, 1954); (2)..."

*- Page 9, line 2: Based on the numerous… here->Based on information collected in field (supplementary Tab. 2), approach (3) was applied here.*

Implemented accordingly.

**Page 6, Lns 37ff:**
"Based on information collected during field surveys (Supplementary Tab. 2) and limited ice dynamics, approach (3) was applied here."

*- Page 9, line 12: Please add some more details on how zones were mapped prior to the 2013 surveys. Does this yield all 5 sites as one of them not measured and some began in 2012 and 2013 according to section 3.2.1?*

Done and yes.

**Page 6, Lns 55ff:**
"Due to field observations and repeated oblique and aerial orthoimagery, the spatio-temporal evolution of the firn thicknesses and extents during and prior to the measured years 2013–2015 could be assessed, and firn compaction assumed to be negligible as a result."

*- Page 9, line 14: Be specific on the displacement, e.g. <xx m/a*

Now implemented accordingly.

**Page 6, Lns 59ff:**
"Ice dynamics were likely negligible for the study glaciers as measured surface displacement rates (in the order of a few $10^0$ m yr$^{-1}$) were always smaller than the resolution of the LiDAR DEMs (several $10^0$ m), and…"

*- Page 9, line 27: The information on the startup of the programmes could have been added in section 2.*

Now we refer to section 3.2.1 in section 2 (see response above).

*- Page 9, line 28: Mostly->Usually*

Done.

*- Page 10, line 14: Could add short details on the stations with reference to the data (e.g. MeteoSwiss?) Use 'and' or 'or' not 'and/or'. Do you mean that you use their results or their methodology, a bit unclear.*

In our opinion, there is no need to give more details on the weather stations which refer to our meteo data used (beyond the scope here). All weather stations are indeed included in the MeteoSwiss network. We therefore now refer to MeteoSwiss in the text. 'and/or' is now replaced by 'or' only.

**Page 7, Lns 18ff:**
"… as well as daily air temperature and precipitation data from nearby MeteoSwiss weather stations. A detailed description of the methodology to infer distributed mass balance is given in Huss et al. (2009) or Sold et al. (2016)."

*- These results would be interesting to compare with the spatial pattern found from the annual/biannual geodetic surveys, has this been compared?*

Yes. Please refer to section 5.2 and Fig. 4 vs. Supplementary Fig. 1.

*- Page 11, Table 3: Text. Divide sentence, is very long and hard to read.*

Table caption is now shortened.

**Page 8, Table caption:**
"Limits of detection for the TLS-derived surface elevation changes ($\sigma\_MSA$) and number of points used for the Multi-Station Adjustment fine registration of consecutive point clouds (*n*) for both observation periods and the surveyed Glacier de Prapio, Glacier du Sex Rouge, St. Annafirn, Schwarzbachfirn, and Pizolgletscher. In addition, the mean ($\mu$), median ($\sim$X), standard deviation ($\sigma$) and interquartile range (iqr) of elevation differences from the comparison of TLS-derived annual surface elevation changes over stable terrain (all in m) are given."

*- Page 12, are not snow patches masked out?*

Snow patches adjacent to the glaciers were, of course, not used for registration of two consecutive LiDAR point clouds. Nevertheless, they appear afterwards on the DEMs of Difference (as for instance on Fig. 3a which we refer to here).

*- Page 15, line 22: can remove 'very small'*

Done.

*- Page 15, line 24: Give results first, before interpreting them for better flow*

Implemented accordingly.

**Page 10, Lns 24ff:**
"Measured mass losses were remarkably stronger for the second time period (–1.65 m w.e. in 2014/15

averaged for the four glaciers measured with both methods compared to –0.59 m w.e. in 2013/14), which agrees well with the different prevailing atmospheric conditions (especially in summer) recorded during the observed years (MeteoSwiss, 2015, 2016).”

*- Page 15, line 25: why not give estimate for geodetic for all five?*

Because we could not validate the TLS-derived geodetic mass balance for Glacier de Prapio and explicitly refer to the glaciers “measured with both methods” here.

*- Page 20, line 7: This can be explained by the higher point density and more complete coverage than for most other glaciers (...)*

Implemented accordingly.

**Page 12, Lns 25ff:**
“This can be attributed to the higher density and more complete coverage of winter and summer point measurements for our study glaciers than for most other glaciers (Supplementary Tab. 2; WGMS, 2013).”

*- Page 20, line 10: Remove With comparatively low uncertainty*

Done.

**Page 12, Lns 29ff:**
“Their very small surface area and the absence or minor fractions of very steep and/or heavily crevassed zones are, of course, optimal preconditions to accurately measure direct glaciological mass balance.”

*- Page 20, lines 11-12: May remove this inserted clause, which we use..., for better flow*

Done.

**Page 13, Lns 1ff:**
“Hence, the quality of both the geodetic mass balances derived by repeated terrestrial LiDAR surveys and the direct glaciological mass balances extrapolated from dense in-situ measurements is very good.”

*- Page 20, line 14: Be specific on what you recommend here, specify your methodological approach*

We now rephrased the sentence into a more general statement.

**Page 13, Lns 6ff:**
“…we recommend the application of terrestrial laser scanning for future mass balance monitoring of very small Alpine glaciers.”

*- Page 20, line 20: Simplify by starting: Significant amounts of fresh snow or remaining firn on the glacier results is more error-prone....*

Done.

**Page 13, Lns 18ff:**
"Significant amounts of fresh snow or firn on the glacier results in more error-prone conversions of TLS-derived volume to mass changes, even more if no additional in-situ measurements of their area fraction and density are performed."

*- Page 20, line 24: over the last years, which years?*

Changed accordingly.

**Page 13, Lns 21ff:**
"On the other hand, from field evidence we know that along with the recorded atmospheric conditions (especially in summer) and the continuously negative mass balance context in the Swiss Alps over the last decade (WGMS, 2012; Huss et al., 2015), the studied very small glaciers hardly exhibit significant ratios of annual to perennial snow and firn anymore."

*- Page 20, line 25: Delete applying our approach*

Done.

*- Page 20, line 28: Instead of 'Following'... Start sentence 'A disadvantage of using the TLS is the....' And refer to reference in parentheses.*

Now implemented accordingly.

**Page 13, Lns 33ff:**
"Disadvantages of using the long-range TLS system and our approach to derive annual surface elevation and geodetic mass changes of very small Alpine glaciers are the high costs for the purchase of the device itself and licenses for the data analysis software provided by the manufacturer, as well as the complex and time-consuming post-processing of the LiDAR data. The required level of expertise and experience with TLS data acquisition and processing is likely higher than for direct glaciological mass balance monitoring (see e.g., Ravanel et al., 2014)."

*- Page 21, line 2: ...very small glaciers, here defined as (<0.5 km2)...*

Implemented accordingly.

**Page 13, Lns 48ff:**
"Despite their global predominance in absolute number, empirical field data on very small glaciers, here defined as being smaller than 0.5 $km^2$, are currently sparse."

*- Page 21, lines 13,23: very dense -> dense*

Done.

*- Page 21, lines 28-32: Unclear sentences and ending, be specific*

Implemented accordingly.

**Page 13, Lns 95ff:**
"Our results show that, under some restrictions, the TLS-based monitoring approach presented in this paper yields accurate results and is therefore suitable for repeated mass balance measurements of very small Alpine glaciers. The most important shortcomings of our approach are related to the abundance of snow and firn at the time of the TLS surveys. They are insignificant in a highly negative mass balance context, as observed for instance for most of our field sites over the last years. Under these circumstances, laborious, time-consuming, and potentially dangerous field measurements may be circumvented and the uncertain spatial inter- and extrapolation of point measurements over the whole glacier surface be avoided."

---

## Author Response (AR2)

**Author's response to the  editor decision of the paper tc-2016-46 submitted to The Cryosphere (Mauro Fischer, corresponding author):**

Dear Editor,

We want to thank you for your positive comments and final report on our paper resubmitted to The Cryosphere.

We answered and commented on all your technical comments/corrections points . –  Editor's comments are formatted in Times 12 italic, our response in Times 12 normal, and the corresponding revised text including information about the **corresponding line numbers in the new TC manuscript version** in Times 10 normal/bold.

~~- *Title: I wonder if the use of "ultra-long-range" is relatively standard, as it seems to me a bit "excessive". May be that using very long range is enough, or at least the text should inform that is a distance that has been very little used in previous research. As the work deal more in validating the measurements rather than explaining the dynamics of the glacier, may be better to include some reference to the validation itself or the comparison to direct glaciological method (just a suggestion)*~~ *P1. Line 51: I would delete the 'even' together with the 'notably' there are bit too many such words.*

 Done.

 *P3 caption fig 1, line 3: insert '(on left side)' after 'study glaciers'*

Now implemented accordingly .

*the manuscript, probably here, a brief description of climate characteristics of the analyzed glaciers (and main differences if exits) and mentioning how was the climate during the two analyzed years compared to long-term climate (last decades) in the Swiss Alps would help to better interpret the presented data on mass balance.* *P4 table 1 line 1: correct 'parafs' to 'parameters'*

Done.

~~Concerning the reviewer's second point, we agree that data about the climate characteristics and variability of the study sites would help to better interpret the presented mass balance data (comparable to, for instance, López-Moreno et al., The Cryosphere 2016). However, we argue that such analyses go beyond the scope this study, which aims at validating TLS-derived annual geodetic mass balances of very small alpine glaciers with direct glaciological mass balances from dense in-situ measurements. We now refer to a new study by Huss and Fischer (2016, Frontiers in Eartch Science) in the revised version of the manuscript (Page 1, Ln 63) which is about the sensitivity of all very small glaciers in Switzerland to climate change. Further, we now also refer to the 2014 and 2015 annual climate bulletins of MeteoSwiss, which, if desired, will help the reader to better link the resulting measured mass balances as well as their regional and interannual variability to the prevailing atmospheric conditions during the observation periods 2013/14-2014/15 (cf. Page 10, Lns 24ff).~~

*- Data and methods: Page 6 line 31: -Which is the consequence of range ambiguity? A slightly expanded explanation (or a reference) might be useful.* *P4 line 27: there is something wrong in this sentence around '...to the importance...'. It says 'pointed out' so the 'to the importance' can refer to 'pointed'. If you leave away the 'out' it may work (or replace 'pointed out' by 'referred'). Does this then still say what you want?*

We rephrased the sentence as follows:

**Page 5, Lns 26ff:**
 "Huss pointed out the remarkable small-scale variability in accumulation and melt processes, and referred to the importance of snow redistribution and the influence of albedo feedback mechanisms on the mass balance of this very small glacier."

*- Page 7. Even if supplementary material inform of the characteristics of the point clouds, I would mention here some numbers about the most usual (or minimum) density of points acquired for this study. - What is an octree filter?* *P5 line 44: can you confirm that it is V 2.1 (in your review response you said 10.1).*

Yes. 10.1 corresponds to the ArcGIS version used, and 2.1 is indeed the version of the RiscanPRO software used here. We're sorry for this error in the review response.

**Page 5, Lns 33ff:**

**Page 6, Lns 5ff:**

P5 line 58: 'were' instead of 'was' ('a number of ... were..')

Done.

**Page 6, Lns 42ff:**

P6 line 44/45: is Sold et al. 2015 the right reference, in the author response you referred to Cuffey and Paterson 2010.

~~Here, we do not refer to winter snow accumulation measurements, but to snow probings performed if there was a significant amount of fresh snow at the time of the annual LiDAR survey in autumn (as mentioned in the text). Autumnal fresh snow covers are, from our experience, usually spatially much more homogeneous than end-of-winter snow accumulation patterns, so a smaller density compared to the winter surveys was enough. We now refer to the recorded median density (as a number of measurements per square kilometer) of autumnal snow probings on the respective glaciers.~~Yes, Sold et al. 2015 is indeed the right reference, we're sorry for not having updated this in the author response.

**Page 6, Lns 74ff:**

-P6 line 61 and also line 62: few 10^0 m/y? why not say 'few metres'? I would think it should probably be 10^(-1) m/y as it says in response. Maybe something went wrong here. check and correct accordingly.

operated reliably and ground motion was prohibited while scanning, …". So our formulation already implies that for instance small instabilities between the tips of the tripod and the bare rock may occur. 10^0 m/y is right, we're sorry for this error in the author response. We now wrote 'few metres'.

-P10 line 25: I would replace 'stronger' with 'higher', mass loss can not be 'strong' but high or low. I guess because it is a negative number you wanted to avoid confusion but you talk about 'mass loss' and if this is 'high' it is more negative. Where comes from that uncertainties of volume change for ice is set to +/- 20 kg m-3?

This is just a conservative estimate for a range in ice density for small mountain glaciers. Now clarified. Now implemented accordingly.